# Active Learning for Parameter Estimation in the Presence of Noise for Linear Models

## Abstract

Parameter estimation is central to scientific inference, yet standard data collection practices, such as random sampling, often yield inefficient or suboptimal results when data are noisy, imbalanced, or expensive to obtain. In such settings, not all samples equally contribute to inference, motivating the need for principled methods to identify and prioritize the most informative data when data are noisy. We propose an active learning method based on Fisher information that quantifies each sample's contribution to the precision of parameter estimates. Unlike prediction performance-driven active learning, our method explicitly targets the improvement of inference precision rather than predictive generalization. By incorporating an adjusted Fisher Information metric, the framework naturally accounts for measurement noise and heteroscedasticity, assigning a higher value to samples that most effectively reduce estimator variance. We provide theoretical guarantees for both linear and logistic regression, demonstrating faster convergence than CoreSet and BAIT approaches, with gains that scale logarithmically with the unlabeled pool size. Extensions to multivariate and non-Gaussian settings further show that parameter-focused active learning offers a principled, efficient strategy for subset selection – prioritizing the most informative observations under realistic, high-noise scientific conditions.

## 1 Introduction

Many problems across science and engineering ultimately reduce to estimating model parameters from data. Examples range from epidemiology (Wang et al., 2020; Dodd et al., 2023; Kraemer et al., 2025), where researchers infer transmission rates from population-level health records, to finance, where risk models depend on precise parameter estimation, to astrophysics, where cosmological parameters are constrained using large-scale sky surveys (Bocquet et al., 2024; Abbott et al., 2022; To et al., 2021). While many of these applications admit linear formulations, modern challenges introduce high dimensionality, data heterogeneity, and measurement noise, making parameter estimation increasingly dependent on the quality and composition of the data subsets used for analysis.

In practice, scientists and engineers rarely have access to perfectly balanced or uniformly informative datasets. Observations can vary dramatically in their relevance to the underlying parameters of interest; some may contain little usable information, while others can substantially reduce uncertainty in key estimates. When labeling or acquiring new measurements is costly (e.g., van der Wal et al., 2021; Nagasubramanian et al., 2021; Zhou et al., 2023), as in many experimental and observational domains, it becomes essential to prioritize which data points to include or label to achieve the most informative subset for modeling (Qi & Baker, 2025). In such cases, the central question shifts from how to estimate parameters given a set of data points to which data points to select or label to best estimate parameters.

This issue is particularly pressing in domains where data collection or annotation is expensive. In speech recognition, manual transcription of audio remains slow and resource-intensive (Dao et al., 2025); in astronomy, millions of detected events exceed the labeling capacity of experts (Settles, 2009). As datasets continue to grow faster than labeling capacity, subset selection and sampling strategies have become critical for efficient and targeted inference.

Existing data valuation and active learning methods offer principled tools for deciding which data are most valuable to label or retain. Diversity-based approaches, such as CoreSet selection (Sener & Savarese, 2017), aim to capture the overall structure of the input space by minimizing redundancy and maximizing coverage. Uncertainty-based methods, by contrast, prioritize samples on which current models are least confident, assuming these points will yield the greatest reduction in predictive error. These strategies have proven effective for tasks centered on improving predictive accuracy and generalization performance.

However, many scientific inference problems focus on parameter estimation rather than a prediction task. In such settings, the most informative data points are those that most reduce the uncertainty of specific parameters, not necessarily those that improve global prediction accuracy. For example, in linear regression, slope estimation depends primarily on data at the extremes of the covariate distribution, while intercept estimation benefits from points near the mean (Harrison & Coles, 2012; Davis et al., 2011; Chongchitnan & Silk, 2021; Heather et al., 2024). Standard uncertainty- or diversity-based methods cannot easily distinguish these parameter-specific contributions, often allocating labeling effort inefficiently. Thus, parameter-focused subset selection requires a distinct criterion that explicitly links data value to inferential goals.

In this paper, we propose a new framework for parameter estimation, designed to identify and select the most informative samples for estimating user-specified parameters. Our method leverages a Fisher-information-based criterion to quantify each data point's contribution to parameter uncertainty reduction. Unlike accuracy-oriented active learning, our approach directly optimizes for information gain about target parameters rather than minimizing global model error. To address practical challenges such as measurement noise and heteroscedasticity, we introduce an adjusted Fisher Information metric that robustly accounts for uncertainty in both covariates and responses. We provide theoretical guarantees in linear and logistic regression settings and empirically demonstrate that our method achieves faster convergence than BAIT and CoreSet baselines, with gains that scale logarithmically with pool size.

**Contributions.** The key contributions of this work are:

- **Formalization of Parameter-subset-focused Fisher Expected Information Objectives in a pool-based streaming active learning setting.** Unlike typical active learning setups that either use a large unlabeled pool or stream 1 point at a time, we operate within a streaming pool framework. We provide a greedy active learning algorithm to sample points that help best estimate parameters of interest

- **Robustness to measurement error:** We extend Fisher Information–based valuation to incorporate measurement noise, enabling robust data selection under realistic, imperfect conditions.

- **Theoretical and empirical guarantees:** We establish optimality results and closed-form expressions for information gain in linear models and demonstrate superior performance over BAIT and CoreSet in both theoretical analysis and experiments.

## 2 Literature Review

While many data valuation techniques focus on improving predictive performance, scientific applications often require a different objective – enhancing the precision of specific parameters. In such cases, conventional valuation methods that prioritize generalization error may be suboptimal. Alternative frameworks that explicitly consider parameter estimation remain an open area of research, bridging the gap between active learning and experimental design. Future work in this domain aims to develop acquisition functions that prioritize data points most informative for hypothesis-driven modeling, moving beyond classification accuracy or regression as the primary metric for data utility.

### 2.1 Data Valuation Methods

Data valuation quantifies the contribution of individual data points to a model's performance, enabling informed decisions about data acquisition, labeling, pruning, and augmentation (Fleckenstein et al., 2023).

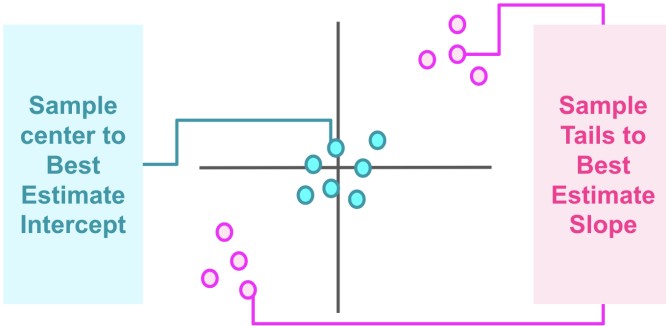

Figure 1: In Linear Regression, sampling around the tails allows better estimation of slope. Sampling around the center allows better estimation of intercept.

This process is particularly relevant in resource-constrained settings, where labeling costs are high or computational efficiency is a priority. By assigning an importance score to each instance, data valuation techniques help optimize training data selection, improve model robustness, and facilitate better generalization (Coleman et al., 2020).

Several approaches have been proposed for assessing data value, often drawing from game theory, influence functions, and information-theoretic principles. These methods differ in how they measure a sample's utility, whether in terms of predictive accuracy, parameter sensitivity, or informativeness for a given learning objective. Below, we outline key frameworks in data valuation.

**Game-Theoretic Approaches** assess data value by modeling the learning process as a cooperative game, where individual data points contribute to a collective objective. The Shapley value, a widely used technique in this category, assigns credit to each sample based on its marginal contribution to model performance when included in various subsets of the training data (Jia et al., 2019). Though theoretically grounded, computing exact Shapley values is combinatorially intractable, leading to the development of approximations such as Monte Carlo sampling and gradient-based estimations (Ghorbani & Zou, 2019). These approximations allow for scalable implementations in high-dimensional settings.

**Influence Functions** provide a principled approach to estimating the impact of a training instance on model parameters and predictions (Hampel, 1974; Cook, 1977; Koh & Liang, 2017). By approximating the effect of perturbing an individual data point, these methods can identify mislabeled or redundant instances and quantify their importance for model generalization. Influence functions are particularly useful in deep learning, where retraining a model from scratch to evaluate the contribution of each sample is infeasible. Recent work has extended these methods to efficiently compute influence scores for large datasets, leveraging curvature approximations of the loss landscape (Guo et al., 2021; Schioppa et al., 2022; Choe et al., 2024).

**Gradient-Based Valuation** techniques assess data importance by analyzing the sensitivity of model parameters to specific training examples. These methods often rely on Fisher Information, which quantifies the amount of information a data point provides about model parameters (Ly et al., 2017). High Fisher Information indicates that a sample plays a critical role in refining parameter estimates, making it particularly valuable for applications where parameter precision is a primary concern. This perspective is closely related to active learning, where data selection is guided by the potential to improve parameter estimation rather than generalization error alone.

## 2.2 Active Learning Methods

Active learning, where the objective is to select the $k$ optimal points from a given pool to achieve the maximal performance achievement, is closely related to data valuation. Since it is computationally intractable to attempt all possible combinations of $k$ points, active learning techniques prioritize the selection of informative instances from an unlabeled dataset to maximize model improvement while minimizing annotation costs. The process of identifying these instances requires quantifying their expected contribution to the model's learning process. Various heuristic strategies have been proposed, each differing in how they estimate the informativeness of a given data point. These strategies often seek to balance exploration, which promotes diversity in the labeled dataset, and exploitation, which targets areas of high uncertainty in the current model. The choice of strategy depends on the specific learning task and the characteristics of the data distribution, making active learning a flexible yet complex framework for optimizing sample efficiency.

To reduce overhead, we only consider single-model methods of active learning, so query-by-committee discussions lie outside the scope of this paper. There are various frameworks for single model instance selection, including the following (Settles, 2009):

**Diversity-based sampling** selects instances that cover a diverse range of the instance space, providing a more comprehensive representation of the dataset. An example of this approach is CoreSet, which applies the facility location problem to choose a diverse set cover from an unlabeled pool of data points (Sener & Savarese, 2017). Other diversity-based approaches aim to increase the representativeness of the sampled data points used in active learning training by performing clustering and selecting the centers of the clusters (Li et al., 2012; Yin et al., 2023; Ienco et al., 2013; Citovsky et al., 2021).

**Uncertainty Sampling** selects instances that the current model finds most uncertain or challenging. These instances are likely to contain valuable information that can improve the model's understanding of the data. This approach creates a heuristic that estimates uncertainty. Some approaches use entropy (Shannon, 1948) or the margin or distance to the decision boundary (Wang & Hua, 2011; Ducoffe & Precioso, 2018) as a heuristic for uncertainty. Bayesian approaches such as BALD aim to approximate parameter posteriors (Houlsby et al., 2011b). Other approaches leverage Fisher Information – a measure of uncertainty with respect to model parameters (discussed further in the next subsection) – to determine the most optimal query (Settles, 2009). These approaches typically aim to reduce overall generalization error and do not necessarily choose the best points for optimizing a specific parameter of interest.

MacKay (1992b) discusses the objective of maximizing expected 'informativeness' of data points within a Bayesian framework by sampling the datapoint with the largest error bars, demonstrating this rule is "D-optimal". In MacKay (1992b) and MacKay (1992a), the information measure is based on entropy.

**Fisher Information** selects instances by quantifying their expected impact on parameter estimation. The Fisher Information Matrix (FIM) measures how much an observation influences the precision of parameter estimates. In the context of active learning, instances with high Fisher Information are prioritized because they provide greater statistical leverage for refining model parameters. Many active learning methods have leveraged Fisher Information to achieve their objective functions. Kirsch & Gal (2022) demonstrates that a common Fisher-based objective of maximizing the expected Fisher information gain (or the BALD algorithm (Houlsby et al., 2011a) is reformulated and leveraged throughout many data subset selection approaches.

Unlike uncertainty sampling, which focuses on decision boundary ambiguity, this method directly targets the informativeness of data in refining the model's internal structure. Recent methods employ tractable approximations of the FIM to select instances efficiently, avoiding the computational burden of explicitly computing second-order derivatives (Ash et al., 2021). This strategy is particularly valuable in scientific applications where parameter estimation is the primary goal rather than overall classification accuracy.

Some algorithms combine uncertainty and diversity-based sampling. For example, BAIT is a state-of-the-art active learning approach that leverages Fisher Information matrices and over-under sampling to tractably choose a subset of points from an unlabeled pool (Ash et al., 2021). Similarly, BatchBALD selects the most mutually informative points within a batch with respect to model parameters based on a greedy approxima-

tion (Kirsch et al., 2019). Active Data Shapely (ADS) instead scores datapoints based on approximations of their shapely values in improving overall model performance (Ghorbani et al., 2022).

These strategies are often designed to enhance the predictive accuracy of a model, which is not necessarily aligned with the goal of maximizing the scientific outcome. In particular, accurately estimating a subset of parameters is not the same objective as minimizing generalization error. Even in some cases, minimizing generalization error can lead to biased parameter estimates (Kosmidis, 2014; Tibshirani & Rosset, 2019). Thus, developing more robust and reliable acquisition functions for the goal of parameter estimation remains an ongoing area of research in active learning.

## 2.3 Active Learning in Linear Settings

Active learning for regression seeks to reduce sample complexity and enhance accurate parameter estimation, particularly in high-dimensional or sparse linear settings. Early work (Cohn et al., 1996; Castro & Nowak, 2008) laid the groundwork for active learning in supervised tasks, but only more recent studies have focused specifically on regression. Riquelme et al. (2017a) introduced TRACE-UCB, an adaptive algorithm for simultaneously estimating multiple linear models under a fixed budget. By balancing the allocation of samples based on unknown noise levels and ensuring well-balanced context distributions, their approach achieves strong performance guarantees even in high-dimensional settings. In the sequential or online regime, Riquelme et al. (2017b) propose a threshold-based method for online active linear regression. This method selectively queries observations whose informativeness, as measured in a transformed space, exceeds a preset threshold, and it naturally extends to sparse high-dimensional settings where only a subset of slopes is nonzero. Chen & Price (2019) further addresses active regression by focusing on sparse linear models. Their linear-sample sparsification technique demonstrates that one can achieve a constant-factor approximation of the least-squares solution with only $O(d)$ labeled samples (where datapoints are in $\mathbb{R}^d$) – improving upon previous approaches based on leverage score sampling that require $O(d \log d)$ labels (Derezinski et al., 2018). Related contributions include Sabato & Munos (2014), who study stream-based active learning for linear regression with random designs and derive lower bounds on sample complexity, and Sugiyama & Nakajima (2009), who propose active learning strategies based on expected error reduction. These works collectively demonstrate that by tailoring the sampling process, it is possible to focus labeling effort on the most informative data points, thereby achieving significant reductions in labeling cost and improved estimation accuracy in linear settings.

Our work is complementary as we focus not on minimizing the empirical or prediction risk, but instead centers on obtaining an accurate estimation of a subset of parameters. Other works in literature have also explored leveraging Fisher Information to best estimate parameters of interest. As shown in Smith et al. (2023) and related work, BALD (Houlsby et al., 2011a) and other Bayesian methods also explicitly target parameter information gain, whereas other approaches like BAIT (Ash et al., 2021), transductive methods, and error-reduction criteria can be prediction-oriented. Our focus is to best estimate user-specified parameters of interest rather than best estimating all parameters or minimizing generalization error.

## 2.4 Connection to Classical Optimal Experimental Design

Our active learning objective is closely related to classical optimal experimental design criteria (Fedorov, 2013; Pukelsheim, 2006). In particular, $D$-optimality minimizes the generalized variance of the full parameter vector by maximizing $\det(\mathcal{I}(\boldsymbol{\theta}))$, $A$-optimality minimizes the average marginal variance by minimizing $\text{tr}(\mathcal{I}(\boldsymbol{\theta})^{-1})$, and $E$-optimality maximizes the smallest eigenvalue of $\mathcal{I}(\boldsymbol{\theta})$; by contrast, $c$-optimality targets a single linear functional $c^\top \boldsymbol{\theta}$ by minimizing $c^\top \mathcal{I}(\boldsymbol{\theta})^{-1} c$. Our method does not aim to optimize the estimation of the entire parameter vector. Instead, it targets a designated subset of scientifically relevant parameters $\boldsymbol{\theta}_s$. Under Assumption 2, the Fisher information is block-diagonal under the partition $(\boldsymbol{\theta}_s, \boldsymbol{\theta} \setminus \boldsymbol{\theta}_s)$, so the asymptotic covariance of the MLE for $\boldsymbol{\theta}_s$ depends only on the Fisher sub-block $\mathcal{I}_{\boldsymbol{\theta}_s \boldsymbol{\theta}_s}$. Consequently, minimizing the aggregate uncertainty in $\boldsymbol{\theta}_s$ corresponds to a *subspace-restricted A-optimality* objective, e.g.,

$$\min \ \text{tr}\big(\mathcal{I}_{\boldsymbol{\theta}_s \boldsymbol{\theta}_s}^{-1}\big) ,$$

rather than global $A$-optimality over all parameters. This viewpoint also clarifies the relation to $c$-optimality: when $|\boldsymbol{\theta}_s| = 1$, the criterion reduces to $c$-optimality for the coordinate direction, whereas for $|\boldsymbol{\theta}_s| > 1$ it can be interpreted as jointly optimizing multiple orthogonal $c$-criteria associated with the components of $\boldsymbol{\theta}_s$. Finally, although our approach is not Bayesian, maximizing Fisher information admits an asymptotic information-theoretic interpretation: under standard regularity conditions, it aligns with maximizing the expected information gain about the parameters induced by the queried covariates.

## 3    Preliminaries and Problem Formulation

### 3.1    Notation

We consider a supervised learning setting with covariates $\mathcal{X} \in \mathbb{R}^{n \times d}$ and responses $\mathcal{Y} \in \mathbb{R}^n$, where $n$ denotes the number of observations, which is assumed to be large. Our focus is on the linear parametric model class $f_{\boldsymbol{\theta}} : \mathcal{X} \to \mathcal{Y}$, where $f_{\boldsymbol{\theta}}$ is parameterized by $\boldsymbol{\theta}$ and specifies a mapping from covariates $\mathbf{x}$ to outcomes $y$. While our analysis is centered on linear models, selected nonlinear extensions are discussed in Section 6.

**Assumption 1.** *Suppose Type-II models, where the parameter dependence only appears in the outcome conditional distribution $P(y, \mathbf{x} \mid \boldsymbol{\theta}) = P(y \mid \mathbf{x}, \boldsymbol{\theta})P(\mathbf{x})$, as opposed to Type-I models, where the parameter dependence appears in both the conditional and marginal distributions.*

Zhang & Oles (2000) argues that active learning is particularly well-suited for type-II models, since the maximum likelihood estimator (MLE) of $\boldsymbol{\theta}$ remains consistent under changes in the covariate marginal distribution $P(\mathbf{x})$. However, the rate of convergence of the estimator is influenced by this distribution. Building on this insight, our strategy selects samples that minimize the variance in parameter estimation, thereby reducing the discrepancy between the true and estimated parameters.

**Streaming Pool Setup.**    We begin with an initial labeled dataset, $\mathcal{D}_L = \{(\mathbf{x}_i, y_i)\}_{i=1}^{N_i}$, where $N_i$ denotes the number of labeled samples available at the outset. The labeling process proceeds over $R$ rounds, reflecting repeated opportunities to acquire additional labels under a limited budget. In each round, the learner is presented with an unlabeled pool $\mathcal{D}_P \subset \mathcal{X}$ of fixed size $m$, drawn i.i.d. from the underlying covariate distribution. This pool represents the set of candidate points that may be labeled in the current round. Because annotation is costly, the learner is allowed to query labels for at most $k \leq m$ points[1], while the remaining $m - k$ points are discarded. This setup reflects practical streaming scenarios where only a small subset of incoming data can be labeled within a fixed budget (e.g., Smailović et al., 2014; Kennamer et al., 2020; Russo et al., 2020; Saran et al., 2023; Capezza et al., 2025).

Formally, the queried set is given by $\mathcal{D}_{\mathcal{K}} = \{(\mathbf{x}_i, y_i)\}_{i \in \mathcal{K}}$, where $\mathcal{K}$ indexes the subset of $k$ chosen instances. The labeled dataset is then updated according to $\mathcal{D}_L \leftarrow \mathcal{D}_L \cup \mathcal{D}_{\mathcal{K}}$, and the model is retrained on the expanded labeled set $\mathcal{D}_L$. The algorithm must decide, at each round, which subset of candidates maximizes the value of the limited labeling budget. After $R$ rounds, the learner has incorporated a total of $kR$ newly labeled samples, in addition to the initial dataset. The result is a model trained on $\mathcal{D}_L$ that reflects the initial data and a curated sequence of queried points.

**Greedy Heuristic** To adapt our setup into the standard pool-based setting, then we assume we have an unlabeled pool of size $(R * \mathcal{D}_P)$. Given the total budget is of size $(R * k)$, then our approach will find the globally optimal set of $(R * k)$ points to sample to best estimate the parameters of interest. Our streaming pool framework means that a locally optimal choice of $k$-points may not be globally optimal, since there could be a streamed pool that has none of the globally optimal points.

**Objective.**    Our objective is to maximize the "*scientific outcome*" by minimizing the variance of estimators for a subset of model parameters. Estimation is carried out via maximization of the log-likelihood function, which serves as the optimization criterion. Formally, let $\boldsymbol{\theta} \in \mathbb{R}^d$ denote the full parameter vector, and let $\boldsymbol{\theta}_s \subseteq \boldsymbol{\theta}$ represent the subset of parameters of primary scientific interest. The goal is to obtain the most accurate estimates of $\boldsymbol{\theta}_s$, in the sense of minimizing their estimation variance using the MLE procedure.

---

[1]We use "labeling" to refer to both categorical and continuous outcomes, depending on the prediction task.

**Assumption 2.** *At the true parameter value $\boldsymbol{\theta}^\star$, the Fisher information matrix with respect to $\boldsymbol{\theta}^\star$ is block-diagonal under the partition $(\boldsymbol{\theta}_s^\star, \boldsymbol{\theta}^\star \setminus \boldsymbol{\theta}_s^\star)$, i.e.,*

$$\mathcal{I}(\boldsymbol{\theta}^\star) = \begin{pmatrix} \mathcal{I}_{\boldsymbol{\theta}_s^\star \boldsymbol{\theta}_s^\star} & \mathbf{0} \\ \mathbf{0} & \mathcal{I}_{\boldsymbol{\theta}^\star \setminus \boldsymbol{\theta}_s^\star, \boldsymbol{\theta}^\star \setminus \boldsymbol{\theta}_s^\star} \end{pmatrix}.$$

### 3.2 Fisher Information Preliminaries

In this section, we introduce the essential preliminaries required for the subsequent analysis. These well-established results can be bypassed by readers already familiar with the material.

The conditional Fisher information, denoted with $\mathcal{I}_\theta(\mathbf{x}, \boldsymbol{\theta}^\star)$, is a measure of the information provided by a data instance $\mathbf{x}$ about a model parameter $\theta \in \boldsymbol{\theta}^\star$. Under certain regularity conditions, the expectation of the score is 0. In our discussion, we assume these regularity conditions hold. Since the expectation of the score is 0, then the conditional Fisher Information is equivalent to the expected value of the squared derivative of the conditional log-likelihood function $\ln P(y \mid \mathbf{x}; f_{\boldsymbol{\theta}})$ with respect to $\theta$, a fixed parameter value

$$\mathcal{I}_\theta(\mathbf{x}, \boldsymbol{\theta}^\star) \equiv \mathbb{E}_{y|\mathbf{x}} \left[ \left( \frac{\partial}{\partial \theta} \ln P(y \mid \mathbf{x}; f_{\boldsymbol{\theta}}) \right)^2 \Bigg|_{\boldsymbol{\theta} = \boldsymbol{\theta}^\star} \right]. \tag{1}$$

The marginal Fisher Information, or simply Fisher Information, denoted by $\mathcal{I}_\theta(\boldsymbol{\theta}^\star)$, is defined as

$$\mathcal{I}_\theta(\boldsymbol{\theta}^\star) \equiv \mathbb{E}_{\mathbf{x}} \mathbb{E}_{y|\mathbf{x}} \left[ \left( \frac{\partial}{\partial \theta} \ln P(y \mid \mathbf{x}; f_{\boldsymbol{\theta}}) \right)^2 \Bigg|_{\boldsymbol{\theta} = \boldsymbol{\theta}^\star} \right]. \tag{2}$$

which quantifies the expected amount of information that the data provide about a model parameter. Intuitively, it measures the sensitivity of the likelihood function to changes in $\theta$: higher values indicate that small perturbations in $\theta$ lead to larger changes in the likelihood, implying that the data are more informative about that parameter. (See Ly et al. (2017) for a more in-depth discussion of Fisher Information.)

The Fisher information has important implications for parameter estimation. If the likelihood function is sharply peaked, indicating high sensitivity to changes in $\theta$, it becomes easier to estimate the parameter from the data. However, if the likelihood function is flat and spread out, more data instances are required to accurately estimate the true value of $\theta^\star$. Generally, under the regularity conditions, for large sample sizes, the difference between the true parameter $\theta^\star$ and the MLE converges in distribution to a normal distribution.

$$\sqrt{n}(\widehat{\theta} - \theta^\star) \xrightarrow{\mathrm{d}} \mathcal{N}(0, \mathcal{I}^{-1}(\theta^\star)), \quad \text{as} \quad n \to \infty,$$

where $n$ is the sample size. Hence,

$$\mathrm{Var}(\widehat{\theta} - \theta^\star) \to \frac{1}{n\mathcal{I}(\theta^\star)}, \quad \text{as} \quad n \to \infty. \tag{3}$$

The validity of this asymptotic normality result relies on certain assumptions and regularity conditions, such as the existence of higher-order moments, regularity of the likelihood function, and correct specification of the underlying statistical model. These conditions ensure the consistency and efficiency of the MLE estimator. Additionally, the Cramer–Rao bound establishes that $\mathcal{I}^{-1}(\theta^\star)$ lower bounds the variance of any unbiased estimator (Rao et al., 1992).

Certain active learning frameworks decompose generalization error into a sum of noise, bias, and model variance. Since their learner cannot optimize the noise or bias, they instead minimize generalization error by minimizing the model variance. Since model variance is inversely proportional to the Fisher Information, these active learning approaches aim to maximize Fisher Information of the parameter space (Settles, 2009).

**Properties of Conditional Fisher Information.** To evaluate Fisher Information in supervised learning settings, we need to know the likelihood $P(y \mid \mathbf{x}; f_{\boldsymbol{\theta}})$. We note that log-likelihood $\ln P(y \mid \mathbf{x}; f_{\boldsymbol{\theta}})$ is used as a loss function in many machine learning problems as well.

**Property 1.** *If data instances $\mathbf{x}_i$ and $\mathbf{x}_j$ are independent, then their conditional Fisher information is additive.*

$$\mathcal{I}(\{\mathbf{x}_i \cup \mathbf{x}_j\}, \theta) = \mathcal{I}(\mathbf{x}_i, \theta) + \mathcal{I}(\mathbf{x}_j, \theta). \tag{4}$$

Proof is in Appendix A. This property implies that the conditional Fisher information for $\theta$ contained in each event is independent of other events.

**Corollary 1.** *Let $\mathbf{x}_1, \ldots, \mathbf{x}_n$ be i.i.d. draws from distribution $q(\mathbf{x})$. Then, under standard regularity conditions,*

$$\mathrm{Var}(\hat{\theta} - \theta^{\star}) \; \to \; \frac{1}{n \, \mathbb{E}_{\mathbf{x} \sim q}[\mathcal{I}(\mathbf{x}, \theta^{\star})]}, \quad as \; n \to \infty. \tag{5}$$

**Lemma 1.** *For a normal likelihood $P(y \mid \mathbf{x}, f_{\boldsymbol{\theta}})$ with homoscedastic variance, the conditional Fisher information for a data instance $\mathbf{x}$ and vector of true parameters $\boldsymbol{\theta}^{\star}$ is:*

$$\mathcal{I}_{\theta}(\mathbf{x}, \boldsymbol{\theta}^{\star}) = c \left[ \frac{\partial f_{\boldsymbol{\theta}}(\mathbf{x})}{\partial \theta} \bigg|_{\boldsymbol{\theta} = \boldsymbol{\theta}^{\star}} \right]^2, \tag{6}$$

*where $c$ is a constant independent of $\mathbf{x}$.*

*Proof.* – Since the likelihood is normal,

$$P(y \mid \mathbf{x}, f_{\boldsymbol{\theta}}) = \frac{1}{\sigma \sqrt{2\pi}} \times \exp \left( \frac{-(y - f_{\boldsymbol{\theta}}(\mathbf{x}))^2}{2\sigma^2} \right)$$

$$\ln P(y \mid \mathbf{x}, f_{\boldsymbol{\theta}}) = \ln \frac{1}{\sigma \sqrt{2\pi}} - \frac{(y - f_{\boldsymbol{\theta}}(\mathbf{x}))^2}{2\sigma^2}$$

We consider the score $S_{\theta}$:

$$S_{\theta} = \frac{\partial}{\partial \theta} \ln(P(y \mid \mathbf{x}, f_{\boldsymbol{\theta}})) = \frac{-1}{2\sigma^2} \times \frac{\partial}{\partial \theta} (y - f_{\boldsymbol{\theta}}(\mathbf{x}))^2 = \frac{(y - f_{\boldsymbol{\theta}}(\mathbf{x}))}{\sigma^2} \times \frac{\partial f_{\boldsymbol{\theta}}(\mathbf{x})}{\partial \theta}$$

Since conditional Fisher Information is the expectation of the score squared under our assumed regularity conditions,

$$\mathcal{I}_{\theta}(\mathbf{x}, \boldsymbol{\theta}^{\star}) = \mathbb{E}_{y \mid \mathbf{x}} \left[ (S_{\theta})^2 |_{\boldsymbol{\theta} = \boldsymbol{\theta}^{\star}} \right]$$

$$= \mathbb{E}_{y \mid \mathbf{x}} \left[ \left( \frac{(y - f_{\boldsymbol{\theta}}(\mathbf{x}))}{\sigma^2} \times \frac{\partial f_{\boldsymbol{\theta}}(\mathbf{x})}{\partial \theta} \right)^2 \bigg|_{\boldsymbol{\theta} = \boldsymbol{\theta}^{\star}} \right]$$

$$= \mathbb{E}_{y \mid \mathbf{x}} \left[ \frac{(y - f_{\boldsymbol{\theta}}(\mathbf{x}))^2}{\sigma^4} \times \left( \frac{\partial f_{\boldsymbol{\theta}}(\mathbf{x})}{\partial \theta} \right)^2 \bigg|_{\boldsymbol{\theta} = \boldsymbol{\theta}^{\star}} \right]$$

$$= \frac{1}{\sigma^4} \left[ \frac{\partial f_{\boldsymbol{\theta}}(\mathbf{x})}{\partial \theta} \right]^2_{\boldsymbol{\theta} = \boldsymbol{\theta}^{\star}} \mathbb{E}_{y \mid \mathbf{x}} \left[ (y - f_{\boldsymbol{\theta}}(\mathbf{x}))^2 \right]$$

$$= \frac{1}{\sigma^2} \left[ \frac{\partial f_{\boldsymbol{\theta}^{\star}}(\mathbf{x})}{\partial \theta} \right]^2_{\boldsymbol{\theta} = \boldsymbol{\theta}^{\star}} \qquad \left( \mathbb{E}_{y \mid \mathbf{x}} \left[ (y - f_{\boldsymbol{\theta}}(\mathbf{x}))^2 \right] = \sigma^2 \right)$$

Due to the homogeneity assumption, $\sigma$ is independent of $\mathbf{x}$ and a constant factor. Thus, the claim holds with $c = \sigma^{-2}$. $\qquad \square$

The additivity property and Corollary 1 hold when $\boldsymbol{\theta}^\star$ is one-dimensional or the parameters are uncorrelated. Specifically, our argument is only valid when the parameters are decoupled, allowing the information from different parameters can be maximized independently. When parameters are not decoupled, the information from different parameters needs to be maximized simultaneously. Hence, one needs to be cautious in generalizing the observations above to models with correlated parameters. By correlation, we mean the non-diagonal elements of the Fisher information matrix are non-zero.

# 4 Method

Given the scientific objective is to best estimate $\boldsymbol{\theta}_s^\star \subseteq \boldsymbol{\theta}^\star$, where $\boldsymbol{\theta}^\star$ is the full set of parameters, our approach employs a Fisher-information-based strategy. By Assumption 1, in the asymptotic regime, we consistently arrive at the same scientific conclusion, irrespective of the random subset chosen for labeling (Zhang & Oles, 2000). This observation substantiates the validity and suitability of employing data valuation methodologies to estimate model parameters.

Our method leverages conditional Fisher Information to pick data points that best reduce the variance of the difference between the true parameters ($\boldsymbol{\theta}_s^\star$) and their respective MLEs ($\widehat{\boldsymbol{\theta}}_s$). Corollary 1 suggests that manipulating $q(\mathbf{X})$ has the potential to amplify or suppress the information gain of our sample (Sourati et al., 2017). Specifically, the optimal sample that best minimizes the variance between the true and estimated parameters is the sample with the maximum total information. There are $\binom{m}{k}$ possible subsets of length $k$ from a pool $\mathcal{D}_P$ of length $m$. The naive approach of manually checking each subset is exponential ($O(\frac{m^k}{k!})$).

Property 1 helps reduce the complexity of identifying the optimal $k$-subset $\mathcal{D}_K$ to $O(m \log m)$. Since information is additive when data observations are independent, then it is sufficient to pick the $k$ data instances with the top information scores. These $k$ points minimize the uncertainty of parameter estimation. This fundamental premise forms the foundation of our work.

Unlike influence function approaches that pick data points to optimize the generalization risk, thereby accounting for the uncertainty in all parameters, our approach allows users to target a parameter subset. Per Assumption 2, users should first decouple their parameters such that parameters in $\boldsymbol{\theta}_s^\star$ are block diagonal.

## 4.1 Motivation

Corollary 1 implies we can adjust the joint covariate distribution of labeled samples such that it reduces the expected estimation $\mathbb{E}\left[|\widehat{\boldsymbol{\theta}}_s - \boldsymbol{\theta}_s^\star|^2\right]$ where $\boldsymbol{\theta}_s^\star \subseteq \boldsymbol{\theta}^\star$ are the true parameter(s) we aim to estimate. Because there is a pool of unlabeled data points $\mathcal{D}_P$ and only a small subset $\mathcal{D}_K$ of this pool can be queried for labeling, users can sample the pool such that the sample distribution approaches the adjusted covariate distribution.

This adjusted distribution is unknown apriori and computationally expensive to compute, as it depends on the distribution of pool data and the unknown true parameters. This problem can be reformulated as finding data points that maximize the conditional Fisher information with respect to the parameters $\boldsymbol{\theta}_s^\star$ that the user aims to estimate. However, we do not claim that these two statements are equivalent. We describe this reformulation and how it relates to our framework before presenting our algorithm.

### 4.1.1 Fisher Information Evaluation

Lemma 1 suggests that when the likelihood $P(y \mid \mathbf{x}; f_{\boldsymbol{\theta}})$ is normal and the noise is independent of $\mathbf{x}$; then the point that maximizes $\left[\frac{\partial f_{\boldsymbol{\theta}}(\mathbf{x})}{\partial \theta}\right]^2_{\boldsymbol{\theta}=\boldsymbol{\theta}^\star}$ also maximizes the conditional Fisher Information for the parameter of interest. This provides us with a fast and scalable measure to compute the conditional Fisher information for large sample sizes. Additionally, Lemma 1 indicates we can compute the conditional Fisher Information of a data instance $\mathbf{x}$ without knowing its corresponding label $y$.

However, conditional Fisher Information $\mathcal{I}_\theta(\mathbf{x}, \boldsymbol{\theta}^\star)$ depends on $\boldsymbol{\theta}^\star$, which is unknown in the situation of parameter estimation. We only have an estimate $\widehat{\boldsymbol{\theta}}$ of the true model parameters $\boldsymbol{\theta}^\star$.

**Proposition 1.** *For a normal likelihood $P(y \mid \mathbf{x}; f_{\boldsymbol{\theta}})$ with homogeneous variance, the conditional Fisher information $\mathcal{I}$ with respect to a fixed parameter $\theta$ is given by*

$$\mathcal{I}_\theta(\mathbf{x}, \boldsymbol{\theta}^\star) = (c + b^2(\mathbf{x})c^2)\left[\left.\frac{\partial f_{\widehat{\boldsymbol{\theta}}}(\mathbf{x})}{\partial \theta}\right|_{\boldsymbol{\theta}=\widehat{\boldsymbol{\theta}}}\right]^2, \tag{7}$$

*where $c$ is a constant and $b(\mathbf{x}) = \mathbb{E}_{y|\mathbf{x}}[y - f_{\widehat{\boldsymbol{\theta}}}(\mathbf{x})] = \mathbb{E}_{y|\mathbf{x}}[f_{\boldsymbol{\theta}^\star}(\mathbf{x}) - f_{\widehat{\boldsymbol{\theta}}}(\mathbf{x})]$.*

Though Equation (7) contains an estimation bias term $b(\mathbf{x})$ that cannot be ignored, in this work, we focus on the first term of the Fisher information. We leave the analysis and estimation of the bias term for future research. We note that if $\widehat{\boldsymbol{\theta}}$ is close to $\boldsymbol{\theta}^\star$ then the bias term should be small.

### 4.1.2 Accounting for the Measurement Error

In practical problems, it is often not possible to measure the outcome variable of interest $y$ perfectly due to various sources of noise and measurement errors. In certain cases, it is possible to estimate or characterize these measurement errors prior to observation (e.g., Creevey et al., 2013; Leung & Bovy, 2019; Anbajagane et al., 2025). Therefore, we assume the measurement error can be estimated even when the labels are unknown.

**Assumption 3.** *The measurement error follows a normal distribution with a known variance ($\sigma_e^2$) and a mean of zero.*

**Theorem 1.** *For a normal likelihood $P(y \mid \mathbf{x}; f_{\boldsymbol{\theta}})$ with homoscedastic variance and known measurement error, the conditional Fisher information is given by*

$$\mathcal{I}_\theta(\mathbf{x}, \boldsymbol{\theta}^\star) = \frac{1}{\sigma^2 + \sigma_e^2(\mathbf{x})}\left[\left.\frac{\partial f_{\boldsymbol{\theta}}(\mathbf{x})}{\partial \theta}\right|_{\boldsymbol{\theta}=\boldsymbol{\theta}^\star}\right]^2. \tag{8}$$

We leverage Assumption 3 and Theorem 1 to introduce the adjusted Fisher Information, denoted by $\mathcal{J}_\theta(\mathbf{x}, \boldsymbol{\theta})$, in Equation (8). This Adjusted conditional Fisher Information metric accounts for the measurement error variance.

$$\mathcal{J}_\theta(\mathbf{x}, \boldsymbol{\theta}) \equiv \frac{1}{\sigma^2 + \sigma_e^2(\mathbf{x})}\left[\left.\frac{\partial f_{\boldsymbol{\theta}}(\mathbf{x})}{\partial \theta}\right|_{\boldsymbol{\theta}}\right]^2. \tag{9}$$

Our method aims to maximize the conditional Fisher information as expressed in Equation (8). Per Assumption 2, it is important to note that this strategy is only valid when the parameters are decoupled, meaning the information from different parameters can be maximized independently. When parameters are not decoupled, the information from different parameters needs to be maximized simultaneously. In the following sections, we will demonstrate the effectiveness of this new measure, the adjusted conditional Fisher information, for parameter estimation.

### 4.1.3 Multivariate Case

Given Assumption 2, the objective function is to choose a data instance $\widetilde{\mathbf{x}} \in \mathcal{D}_P$,

$$\widetilde{\mathbf{x}} = \arg\max_{\mathbf{x}} \operatorname{tr}\big(\mathcal{I}_{\boldsymbol{\theta}_s}(\mathbf{x}; \boldsymbol{\theta})\big) = \arg\max_{\mathbf{x}} \mathbb{E}_{y|\mathbf{x},\boldsymbol{\theta}}\left[\|\nabla_{\boldsymbol{\theta}_s} \log p(y \mid \mathbf{x}; \boldsymbol{\theta})\|_2^2\right].$$

Thus, we can extend Equation (9) such that $\mathcal{J}_{\boldsymbol{\theta}_s}(\mathbf{x}, \boldsymbol{\theta}^\star)$ reflects the information provided by a data instance about a set of model parameters $\boldsymbol{\theta}_s$. The aforementioned properties also apply to the multivariate case,

$$\mathcal{J}_{\boldsymbol{\theta}_s}(\mathbf{x}, \boldsymbol{\theta}) = \sum_{\theta_s \in \boldsymbol{\theta}_s} \frac{1}{\sigma^2 + \sigma_e^2(\mathbf{x})}\left[\left.\frac{\partial f_{\boldsymbol{\theta}}(\mathbf{x})}{\partial \theta_s}\right|_{\boldsymbol{\theta}}\right]^2. \tag{10}$$

## 4.2 Our Algorithm

We present our strategy below. Each round, we assume we are given a new pool of unlabeled data points $\mathcal{D}_P$, following from a streaming setup. We also assume there exists a set of previously labeled points $\mathcal{D}_L$ and current estimated parameters $\widehat{\boldsymbol{\theta}}$ for the parametric model class $f_{\boldsymbol{\theta}} : \mathcal{X} \to \mathcal{Y}$. Note $\widehat{\boldsymbol{\theta}}$ is estimated using the currently labeled data points $\mathcal{D}_L$.

As outlined in Algorithm 1, each round we select $k$ data points from $\mathcal{D}_P$ that maximize information for the parameters we wish to estimate ($\boldsymbol{\theta}_s$). To determine those $k$ data points ($\mathcal{D}_K$), we perform the following:

1. **Information Evaluation**: We evaluate the information content of each pool event in estimating parameter(s) $\boldsymbol{\theta}_s$. In the equation below, $\widehat{\sigma}^2$ represents the sample variance of the labeled dataset $\mathcal{D}_L$. Thus, $\forall \ \mathbf{x}_i \in \mathcal{D}_P$

$$\mathcal{J}_{\boldsymbol{\theta}}(\mathbf{x}_i, \widehat{\boldsymbol{\theta}}) = \sum_{\theta_s \in \boldsymbol{\theta}_s} \frac{1}{\widehat{\sigma}^2 + \sigma_e^2(\mathbf{x}_i)} \left[ \frac{\partial f_{\boldsymbol{\theta}}(\mathbf{x}_i)}{\partial \theta_s} \bigg|_{\boldsymbol{\theta}=\widehat{\boldsymbol{\theta}}} \right]^2 .$$

2. **Sorting**: We sort the data points by their adjusted fisher information $\mathcal{J}_{\boldsymbol{\theta}_s}(.)$.

3. **Selection**: We select the top $k$ data points with maximum $\mathcal{J}_{\boldsymbol{\theta}_s}(.)$ and add those to $\mathcal{D}_L$, the labeled dataset, to leverage in the next round.

---

**Algorithm 1** Each Round $r \in [R]$

---

**Require:** $f_{\boldsymbol{\theta}}, \mathcal{D}_P, \mathcal{D}_L, \boldsymbol{\theta}_s, \widehat{\boldsymbol{\theta}}, k, \sigma_e$

$\triangleright$ (1) Information Evaluation
$\widehat{\sigma}^2 \leftarrow \sum (y - f_{\boldsymbol{\theta}}(\mathbf{x})|_{\boldsymbol{\theta}=\widehat{\boldsymbol{\theta}}})^2 / n \ \ \forall \ \{\mathbf{x}, y\} \in \mathcal{D}_L$

$\mathcal{J}_{\boldsymbol{\theta}_s}(\mathbf{x}, \widehat{\boldsymbol{\theta}}) \leftarrow \frac{1}{(\widehat{\sigma}^2 + \sigma_e^2(\mathbf{x}))} \sum_{\theta \in \boldsymbol{\theta}_s} \left[ \frac{\partial f_{\boldsymbol{\theta}}(\mathbf{x})}{\partial \theta} \big|_{\boldsymbol{\theta}=\widehat{\boldsymbol{\theta}}} \right]^2 \ \ \forall \ \mathbf{x} \in \mathcal{D}_P$

$\triangleright$ (2) Sort points in descending order by their Adjusted Information $\mathcal{J}_{\boldsymbol{\theta}_s}(\mathbf{x}, \widehat{\boldsymbol{\theta}})$
$\text{Sort}(\mathcal{D}_P, \mathcal{J}_{\boldsymbol{\theta}_s}) \leftarrow (\mathbf{x}_1, ..., \mathbf{x}_m)$ \hfill $\triangleright \ \mathcal{J}_{\boldsymbol{\theta}_s}(\mathbf{x}_1, \widehat{\boldsymbol{\theta}}) \geq ... \geq \mathcal{J}_{\boldsymbol{\theta}_s}(\mathbf{x}_m, \widehat{\boldsymbol{\theta}})$

$\triangleright$ (3) Pick k points with max Adjusted Information
$\mathcal{D}_K \leftarrow \{\text{Sort}(\mathcal{D}_P, \mathcal{J}_{\boldsymbol{\theta}_s})[i] \ : \ \forall \ i \in [1, k]\}$
$\mathcal{D}_L \leftarrow \mathcal{D}_L \cup \{(\mathbf{x}^*, y^*) \ : \ \mathbf{x}^* \in \mathcal{D}_K, \ y^* = \text{label}(x^*)\}$ \hfill $\triangleright$ Add labeled point to $\mathcal{D}_L$
**return** $\mathcal{D}_L$

---

We note that while our algorithm supports selecting multiple points per round, the acquisition function is additive and evaluates points independently. Thus, it does not explicitly penalize redundancy within a batch. This design choice follows from the additivity of Fisher information under independent sampling and aligns with our goal of minimizing marginal parameter variance rather than maximizing joint information gain.

It is important to note our acquisition function is equivalent to maximizing the Expected Information Gain, and in Bayesian models, our method matches the BALD criteria (Kirsch & Gal, 2022; Houlsby et al., 2011a). Our contribution is not to introduce a new information measure, but to (i) isolate the parameter-subspace version of this objective, (ii) analyze it explicitly in linear and generalized linear models, and (iii) study its behavior under heteroscedastic measurement error in a streaming pool regime.

## 4.3 Evaluation: Gain Factor

Ultimately, our objective is to minimize the variance of the difference between the estimated and true parameters of interest. We can quantify the performance of a given sampling strategy by comparing $\mathbb{E}\left[|\widehat{\boldsymbol{\theta}}_s - \boldsymbol{\theta}_s^\star|^2\right]$ from the sampling regime to that from random sampling. We note that $\mathbb{E}\left[|\widehat{\boldsymbol{\theta}}_s - \boldsymbol{\theta}_s^\star|^2\right]$ is equivalent to

$\text{Var}(\widehat{\boldsymbol{\theta}}_s - \boldsymbol{\theta}_s^\star)$ when $\widehat{\boldsymbol{\theta}}_s$ is an unbiased estimate of $\boldsymbol{\theta}_s^\star$. A sampling strategy outperforms random sampling if it yields a lower variance than the random sampling strategy. By Equation (3), this is asymptotically equivalent to the given sampling strategy having higher information than the random strategy. We define **Gain Ratio**, which measures how well a given sampling algorithm performed compared to random sampling, as

$$g_{\boldsymbol{\theta}_s} \equiv \frac{\mathbb{E}_{q(\mathbf{X})} \left\| \boldsymbol{\theta}_s^\star - \widehat{\boldsymbol{\theta}}_{s,\text{random}} \right\|^2}{\mathbb{E}_{q(\widetilde{\mathbf{X}})} \left\| \boldsymbol{\theta}_s^\star - \widehat{\boldsymbol{\theta}}_{s,\text{algo}} \right\|^2}. \tag{11}$$

If an algorithm possesses a higher gain ratio than a different algorithm, then it better achieves the objective. Asymptotically, if we apply Corollary 1, **Gain Ratio** becomes

$$g_{\boldsymbol{\theta}_s} \to \frac{\mathbb{E}_{q(\widetilde{\mathbf{X}})}[\mathcal{I}_{\boldsymbol{\theta}_s}(\widetilde{\mathbf{X}}, \boldsymbol{\theta}_s^\star)]}{\mathbb{E}_{q(\mathbf{X})}[\mathcal{I}_{\boldsymbol{\theta}_s}(\mathbf{X}, \boldsymbol{\theta}_s^\star)]}, \quad \text{as} \quad n \to \infty. \tag{12}$$

$\boldsymbol{\theta}_s$ represents the parameter(s) of scientific interest. $q(\widetilde{\mathbf{X}})$ is the joint distribution of the labeled sample via an active learning algorithm and $q(\mathbf{X})$ is the joint distribution of the random sample. The gain ratio quantifies how much one can expect to gain by switching from a passive algorithm to an active learning algorithm.

## 5 Linear Regression

We investigate the characteristics of Fisher information in a linear regression setting. Our aim is to gain insights into how our strategy, built on the Fisher information, works, understand its properties, and quantify the potential improvement in constraining power compared to random sampling. This example provides a theoretical justification for the proposed method and sheds light on its properties.

### 5.1 Gain Ratio in Univariate Linear Regression

Let $x$ be drawn from a normal distribution with mean zero and $s^2$ sample variance, $x \sim \mathcal{N}(0, s^2)$. The generative model is a linear model with $y = \beta^\star x + \alpha^\star + \varepsilon$, where $\varepsilon$ is normally distributed with mean zero and variance $\sigma^2$, $\mathbb{E}[x] = \mu$, and $\mathbb{E}\left[(x - \mu)^2\right] = s^2$. Our scientific goal is to estimate the parameters $\alpha^\star$ and $\beta^\star$ as precisely as possible.

The problem setup is the same as described in our notation, with $\boldsymbol{\theta}^\star = \begin{bmatrix} \alpha^\star \\ \beta^\star \end{bmatrix}$.

The conditional Fisher information for estimating $\alpha^\star$ is

$$\mathcal{I}_{\alpha^\star}(x, \boldsymbol{\theta}^\star) = \frac{1}{\sigma^2}. \tag{13}$$

This indicates that the conditional Fisher information about the intercept, $\alpha^\star$, for all data points is equal. Thus, manipulating the distribution of labeled sample $q(\mathbf{x})$ does not lead to any additional information gain. Therefore, an active learning strategy based solely on the conditional Fisher information cannot lead to an improvement when estimating the intercept $\alpha^\star$. In contrast, the conditional Fisher information of $\beta^\star$ at point $x$ is

$$\mathcal{I}_{\beta^\star}(x, \boldsymbol{\theta}^\star) = \frac{(x - \mu)^2}{\sigma^2}, \tag{14}$$

which is a function of $x$. Consequently, the most informative data points in the linear setting for estimating $\beta^\star$ have the highest $(x - \mu)^2$. Thus, to maximize the Fisher Information per labeling iteration, it is more optimal to choose the $k$ data points with the highest $(x_i - \mu)^2$. This translates to selecting the $k$ data points located at the tails of the distribution, farthest from the mean. These selected data instances are rare examples that are not representative of the population but provide the most useful information about slope.

We also note that the standard error of the estimated slope is

$$\text{SE}(\widehat{\beta}) = \sqrt{\frac{\sigma^2}{n\text{Var}(\widetilde{X})}}. \tag{15}$$

Here, $\widetilde{X}$ represents the labeled sample and $\text{Var}(\widetilde{X})$ represents the variance of the labeled sample. Under the random sampling strategy (passive learning), $\text{Var}(\widetilde{X}) = \text{Var}(X) = s^2$, where $X \sim p$. In this case, $\text{SE}(\widehat{\beta}) = \sqrt{\frac{\sigma^2}{ns^2}} = \frac{\sigma}{s\sqrt{n}}$. This means the random sampling strategy yields a convergence rate of $\propto 1/\sqrt{n}$. Under our strategy, $\widetilde{X} \sim q$ and $q$ can be different than $p$. The **Gain Ratio** in univariate linear regression settings is

$$g_\beta \equiv \frac{\mathbb{E}_{X \sim p}||\beta^\star - \widehat{\beta}_{\text{random}}||^2}{\mathbb{E}_{\widetilde{X} \sim q}||\beta^\star - \widehat{\beta}_{\text{algo}}||^2} \approx \frac{\left(\text{SE}(\widehat{\beta}_{\text{random}})\right)^2}{\left(\text{SE}(\widehat{\beta}_{\text{algo}})\right)^2} \approx \frac{\text{Var}(\widetilde{X})}{\text{Var}(X)} \tag{16}$$

### 5.1.1 Distribution of Algorithm 1's labeled points

To estimate $\beta^\star$, each round, we select the top $k$ points with maximum $\mathcal{J}_\beta(.)$. In the linear regression case, $f_{\boldsymbol{\theta}}$ is the linear model $\widehat{y} = \widehat{\beta}x + \widehat{\alpha}$. Hence, we have

$$\mathcal{J}_\beta(x_i, \widehat{\boldsymbol{\theta}}) = \frac{1}{\widehat{\sigma}^2 + \sigma_e^2(x_i)} \left[ \frac{\partial f_{\boldsymbol{\theta}}(x_i)}{\partial \beta} \bigg|_{\boldsymbol{\theta}=\widehat{\boldsymbol{\theta}}} \right]^2 = \frac{1}{\widehat{\sigma}^2 + \sigma_e^2(x_i)} \left[ \frac{\partial(\beta(x_i - \widehat{\mu}) + \alpha)}{\partial \beta} \bigg|_{\beta=\widehat{\beta}} \right]^2 = \frac{(x_i - \widehat{\mu})^2}{\widehat{\sigma}^2 + \sigma_e^2(x_i)}$$

Thus, in linear regression, our strategy selects the top $k$ points with the highest $(x_i - \widehat{\mu})^2$ where $\widehat{\mu}$ is the empirical mean.

### 5.1.2 Relation between Gain Ratio and Pool Size

In this section, without loss of generality and for notational convenience, we assume $\mu = 0$.

**Theorem 2.** *Let $\mathcal{D}_P = \{x_1, \ldots, x_m\}$ be samples drawn from a normal distribution with mean zero and $s^2$ variance, i.e $\forall i \in [1, m]$. $x_i \sim \mathcal{N}(0, s^2)$. Let $\widetilde{X} = \{x_{(m-k-1)}, \ldots, x_{(m)}\}$ be the $k$ points chosen under Algorithm 1 for sampling. (Note: $\widetilde{X} \subseteq \mathcal{D}_P$ and $x_{(i)}$ is the ith order statistic and $x_{(i)} \leq x_{(i+1)}$). Then, the probability distribution of $\widetilde{x} \in \widetilde{X}$ is*

$$P(\widetilde{x}, m, k) = \frac{m!}{(m-k)!} \times \left[ \prod_{i=m-k+1}^{m} \frac{1}{s\sqrt{2\pi}} \exp\left( \frac{-(x_{(i)})^2}{2s^2} \right) \right] \times \left[ 2\Phi\left( \frac{|x_{(m-k+1)}|}{s} \right) - 1 \right]^{m-k} \tag{17}$$

*where $\Phi\left( \frac{x}{s} \right) = \frac{1}{2} \left[ 1 + \text{erf}(\frac{x}{s\sqrt{2}}) \right]$ is the CDF of $\mathcal{N}(0, s^2)$.*

Theorem 2 means the gain ratio of our algorithm with respect to random sampling is positively correlated with both $m$ and $k$. When $k = 1$ (choose 1 point per pool), then $\widetilde{X} = \{x_{(m)}\}$, where $x_{(m)}$ is the maximal order statistic. Then, the PDF of $\widetilde{x} := x_{(m)}$ simplifies to the following form:

$$P(\widetilde{x}, m, k = 1) = m \times \frac{1}{s\sqrt{2\pi}} \exp\left( \frac{-\widetilde{x}^2}{2s^2} \right) \times \left[ 2\Phi\left( \frac{|\widetilde{x}|}{s} \right) - 1 \right]^{m-1} \tag{18}$$

Figure 2(a) shows the relationship between the gain factor of our strategy and the pool size, when the budget $k = 1$. To estimate $\text{Var}(\widetilde{X}, m) = \mathbb{E}_{\widetilde{X} \sim P(.; m, k=1)} \left[ \widetilde{X}^2 \right]$ we sample from distribution of Equation (18). Similarly, Figure 2(b) shows the relationship between the gain factor of our strategy and the budget size, when pool size $m = 10,000$. We estimate $\mathbb{E}_{\widetilde{X} \sim P(.; m=10000, k)} \left[ \widetilde{X}^2 \right]$ by sampling from Equation (21).

**Interpretation.** The findings derived from Figure 2(a) indicate a logarithmic increase in the gain factor as the pool size expands, suggesting that larger pool sizes can yield more accurate estimation with smaller labeled samples. For example, with a pool size of approximately 200, our approach can achieve a comparable

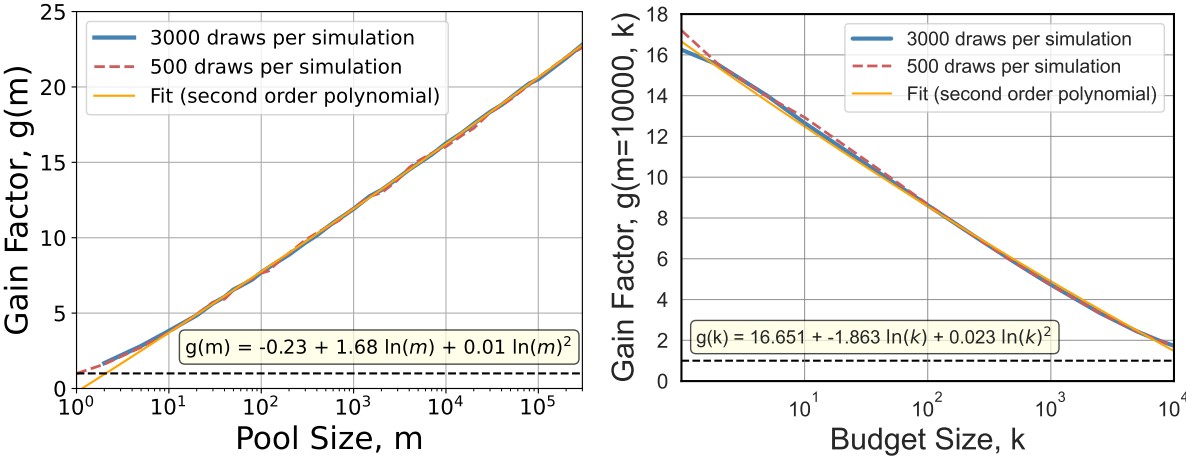

Figure 2: For both plots, we calculate the gain factor in estimating the slope in a linear regression setting. The blue solid and red dashed lines represent independent simulations, each with 3000 and 500 draws, respectively. The yellow line is an empirical second-order polynomial fit to the blue curve. **Left:** The figure shows the gain factor as a logarithmic function of pool size, with a fixed budget of $k = 1$. The fit is only valid for pool sizes spanning from 10 to $10^5$. Our findings indicate a logarithmic **increase** in the gain factor with respect to the pool size for this setting. **Right:** Gain factor in estimating the slope in a linear regression setting. The figure shows the gain factor as a logarithmic function of budget size, with a fixed pool size of $m = 10,000$. Our findings indicate a logarithmic **decrease** in the gain factor with respect to the budget size for this setting.

confidence interval on the slope with only one-third of the labeled data required. However, to attain a factor of four improvement in confidence intervals, a larger pool size of around 10000 becomes necessary. Conversely, Figure 2(b) indicates a logarithmic decrease in the gain factor as the budget increases. This makes sense since increased budget means our distribution of selected points grows closer to the underlying pool distribution (since a larger portion of the pool is selected).

In univariate case, our algorithm coincides with active learning algorithms that choose samples with the highest level of uncertainty in their predictions. We show this more formally in the appendices.

### 5.1.3 Experimental Setup

To evaluate our approach, we adapt other active learning approaches in literature to estimate the parameters of interest in a streaming pool setup. Our features $X \sim \mathcal{N}(0, 1)$ and redefine a generator model $g$ such that $Y = g(X)$. Each approach begins with the same set of randomly selected $s = 10$ initial data points drawn from the underlying distribution. We design a simulation where in each iteration we generate a pool of data points from an underlying true distribution. We then apply each algorithm to select $k = 1$ data points from the same pool to label and add to the algorithm's labeled set. We record the difference between the model and true parameters and repeat this procedure for $n$ realizations to construct a distribution. We then compare the variance across all the $n$ realizations of the difference between the estimated and true model parameters.

Specifically, we compare our approach against passive learning (random sampling), the CoreSet Algorithm, and the BAIT Algorithm. CoreSet employs a diversity-based approach and selects points farthest from the nearest labeled datapoints (Sener & Savarese, 2017). BAIT leverages Fisher Information and over-under sampling to select a heuristic estimate for the most informative datapoints (Ash et al., 2021). Both CoreSet and BAIT demonstrate state-of-the-art results on multiple benchmarks. We thus apply these algorithms to the parameter estimation in a streaming pool problem and compare their results.

### 5.1.4 Univariate Setting Without Measurement Error

We define $y = g(x) = \beta^\star x + \alpha^\star + \varepsilon$, where $\varepsilon \sim \mathcal{N}(0,1)$ and $\alpha^\star = 0$, $\beta^\star = 1$. We seek to estimate $\beta^\star$. As shown in Figure 3, our sampling strategy results in a faster convergence compared to other approaches. We also see in Figure 4 that the gap between our approach and other algorithms widens with larger pool sizes, matching the theoretical analysis in Section 5.1.2.

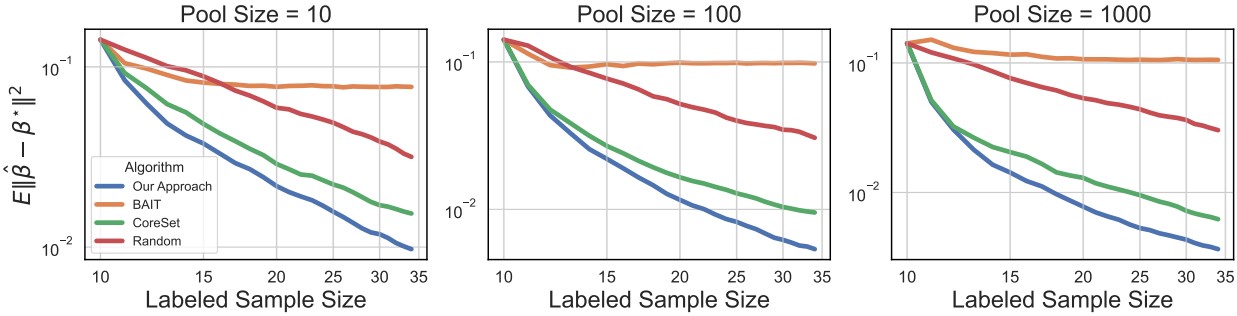

Figure 3: Given the generator function is $y = g(x) = \beta^\star x + \alpha^\star + \varepsilon$, we estimate $\beta^\star$. The Variance is calculated over $n = 1000$ realizations and plotted against $|\mathcal{D}_L|$, the labeled sample size. Each algorithm is initialized with the same $s = 10$ randomly selected points. Each round, each algorithm receives the same $\mathcal{D}_P$ but chooses its own set of $k = 1$ points to label. In this plot, **lower is better**. This plot has been updated so BAIT trains on original samples, rather than decoupled samples.

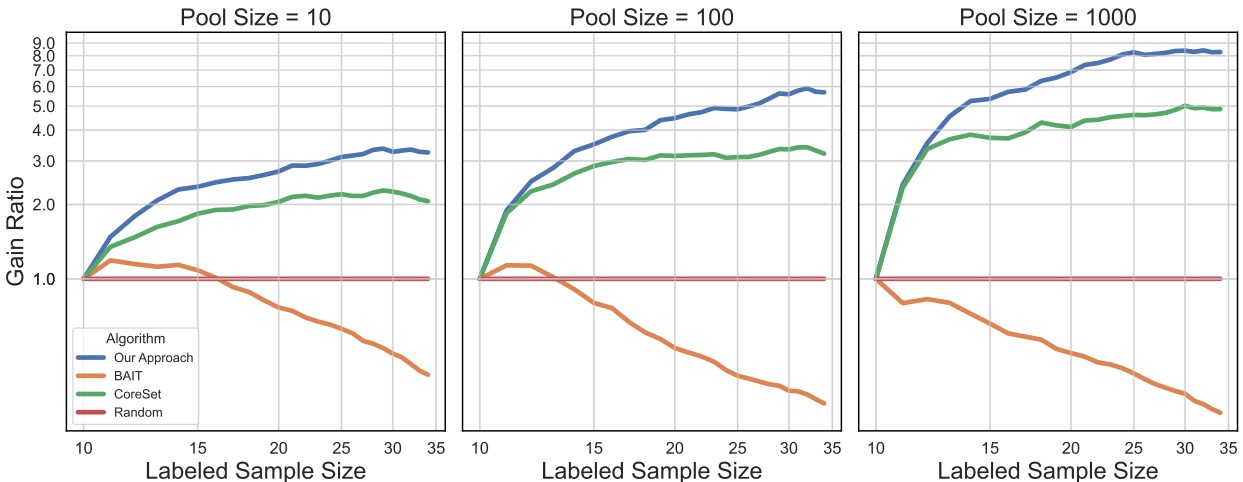

Figure 4: We repeat the experiment in Figure 3, but instead plot **Gain Ratio** (Equation 12) over $n = 1000$ realizations as a function of $|\mathcal{D}_L|$, the labeled sample size. In this plot, **higher is better**. This plot has been updated so BAIT trains on original samples, rather than decoupled samples.

**Ablation Study for Assumption 2** To clarify the importance of Assumption 2, we conduct an ablation study. In this experiment, to more clearly see the impact of this assumption, we draw $X \sim \mathcal{N}(2,1)$. As seen in Figures 5 and 6, our approach with no decoupling performs worse than our approach with decoupling and the CoreSet algorithm. This is because if we do not decouple the parameters by subtracting the estimated mean from the unlabeled pool, then we select points from only 1 tail of the distributions. However, by decoupling our model parameters, we shift the distribution of pool points, allowing us to select points from both tails and thus better estimate $\beta^\star$.

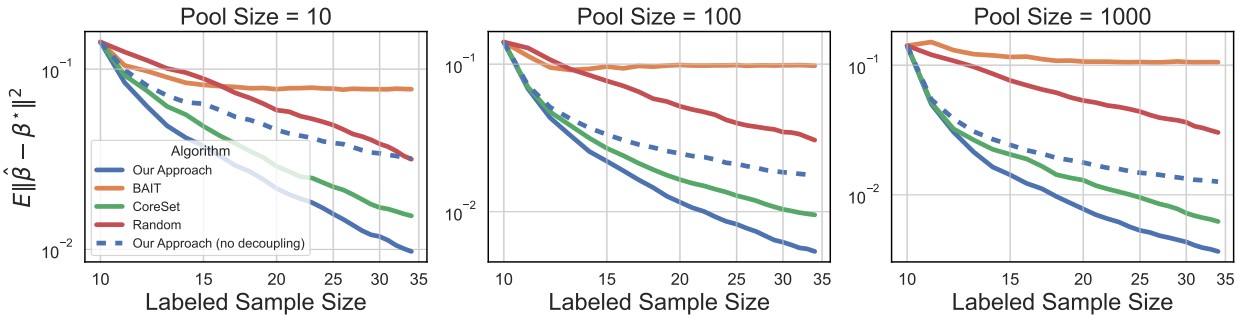

Figure 5: Given the generator function is $y = g(x) = \beta^\star x + \alpha^\star + \varepsilon$ and that $X \sim \mathcal{N}(2, 1)$, we estimate $\beta^\star$. The Variance is calculated over $n = 1000$ realizations and plotted against $|\mathcal{D}_L|$, the labeled sample size. Each algorithm is initialized with the same $s = 10$ randomly selected points. Each round, each algorithm receives the same $\mathcal{D}_P$ but chooses its own set of $k = 1$ points to label. In this plot, **lower is better**.

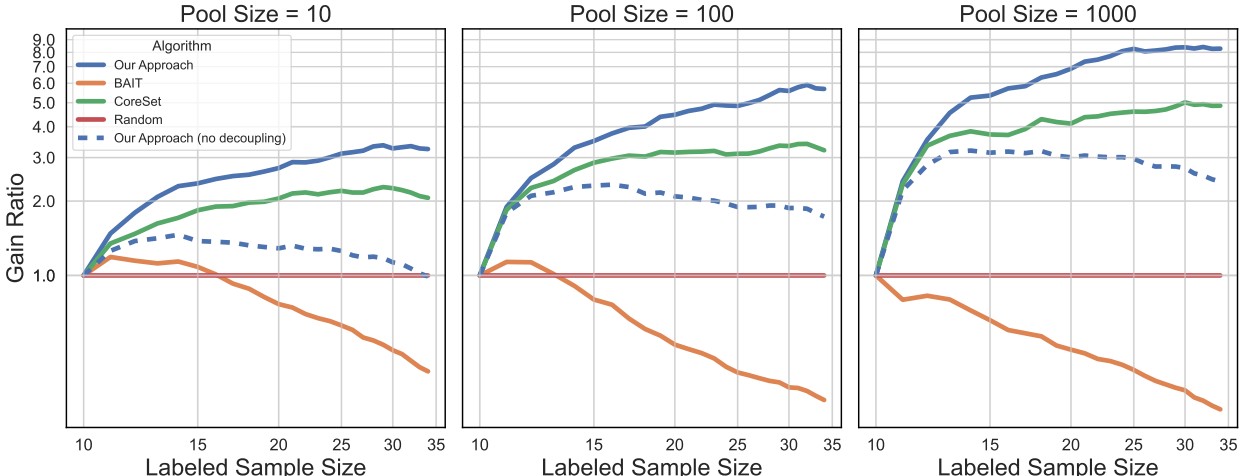

Figure 6: We repeat the experiment in Figure 5, but instead plot **Gain Ratio** (Equation 12) over $n = 1000$ realizations as a function of $|\mathcal{D}_L|$, the labeled sample size. In this plot, **higher is better**.

### 5.1.5 Univariate Setting With Measurement Error

Figure 7 demonstrates the difference between the distribution of sampled points when we exclude and include measurement error. As seen in Figure 7, adding measurement error shifts the distribution of sampled points in the direction of the measurement error. Specifically, in the presence of measurement error, we see more uneven peaks in our approach (with narrower peaks as pool size increases).

We add measurement error $\delta \sim N(0, \sigma_e^2)$ to our generator model such that $y = \beta^\star x + \alpha^\star + \varepsilon + \delta$, where measurement error $\delta \sim \mathcal{N}(0, \sigma_e^2)$ and $\sigma_e \sim \text{LogNormal}(1, 2)$. As seen in Figures 8 and 9, accounting for measurement error worsens performance across the board as the variance in the difference between the estimated $\widehat{\beta}$ and true $\beta^\star$ is higher than the case without measurement error. This is expected, since estimating model parameters with measurement errors present is a more challenging task.

Furthermore, in this experiment, we include an *ablation* comparing the performance of our algorithm that accounts for measurement error against our algorithm that assumes no measurement error exists. For a fair comparison, we modify BAIT to estimate variance accounting for measurement error. As seen in Figures 8 and 9, our approach benefits greatly from accounting for measurement error.

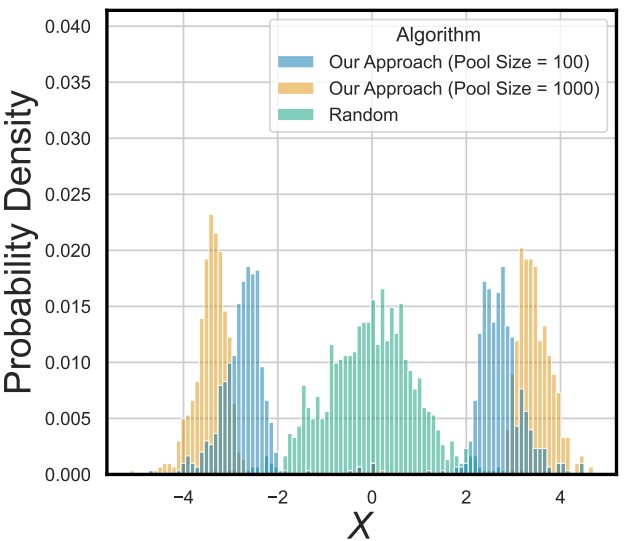 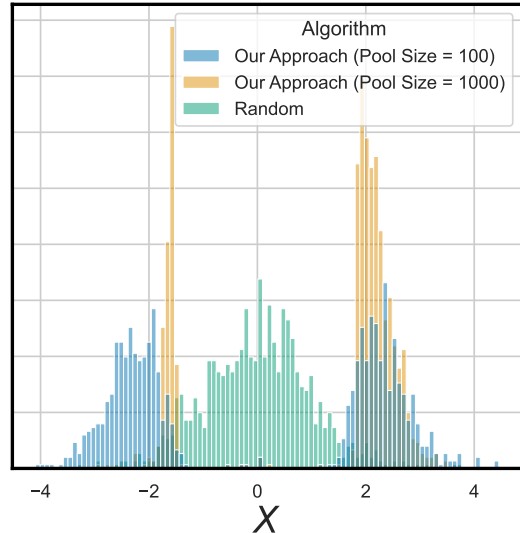

Figure 7: In both plots we sample and plot the $i = 1000$ additional points chosen by each algorithm, given $s = 10$ initial random datapoints. We set $\alpha^\star = 0$ and $\beta^\star = 1$. We seek to estimate $\beta^\star$. **Left:** The generator function is $y = \beta^* x + \alpha^* + \varepsilon$ (where no measurement error is present). **Right:** The generator function is $y = \beta^* x + \alpha^* + \varepsilon + \delta$, where measurement error $\delta \sim \mathcal{N}(0, \sigma_e^2)$ and $\sigma_e \sim \text{LogNormal}(1, 2)$.

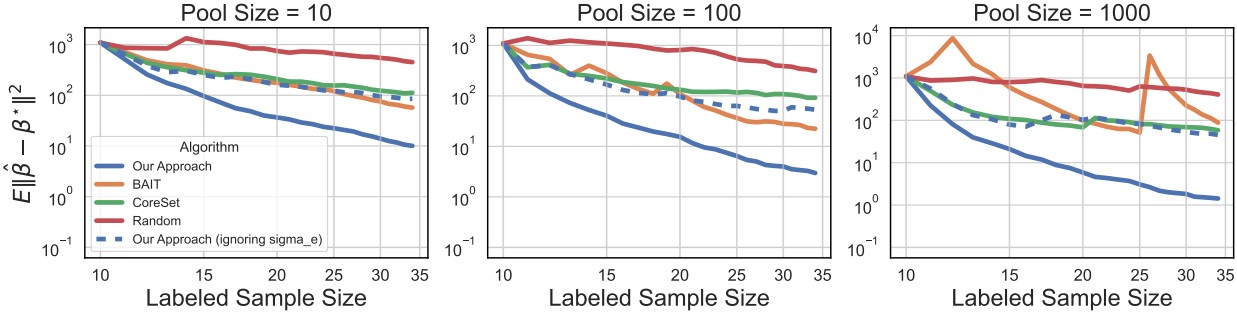

Figure 8: The generator function is $y = \beta^* x + \alpha^* + \varepsilon + \delta$, where measurement error $\delta \sim \mathcal{N}(0, \sigma_e^2)$ and $\sigma_e \sim \text{LogNormal}(1, 2)$. We estimate slope $(\beta^\star)$ in univariate linear regression with measurement error $\sim \mathcal{N}(0, \sigma_e^2)$. The variance is calculated over $n = 2500$ realizations and plotted against $|\mathcal{D}_L|$, the labeled sample size. Each algorithm is initialized with the same $s = 10$ randomly selected points. Each round, each algorithm receives the same $\mathcal{D}_P$ but chooses $k = 1$ points to label. In this plot, **lower is better**.

## 5.2 Multivariate Linear Regression

We can apply Equation 10 to multivariate linear regression as follows:

$$\mathcal{J}_{\boldsymbol{\theta}_s}(\mathbf{x}, \boldsymbol{\theta}) = \sum_{\theta_s \in \boldsymbol{\theta}_s} \frac{1}{\sigma^2 + \sigma_e^2(\mathbf{x})} \left[ \frac{\partial f_{\boldsymbol{\theta}}(\mathbf{x})}{\partial \theta_s} \bigg|_{\boldsymbol{\theta}} \right]^2 = \sum_{\theta_s \in \boldsymbol{\theta}_s} \frac{(x_s - \bar{x}_s)^2}{\sigma^2 + \sigma_e^2(\mathbf{x})} \tag{19}$$

The empirical mean $\bar{x}_s = \sum_{i=1}^{|\mathcal{D}_L|} x_{s,i} / |\mathcal{D}_L|$ is subtracted from the pool data to decouple the normalization error from the slopes and make the Fisher information matrix diagonal.

Thus, similar to the univariate case, our algorithm selects the points with the highest squared distance from the mean in the parameter space of interest. This means the measure of the sampled points essentially

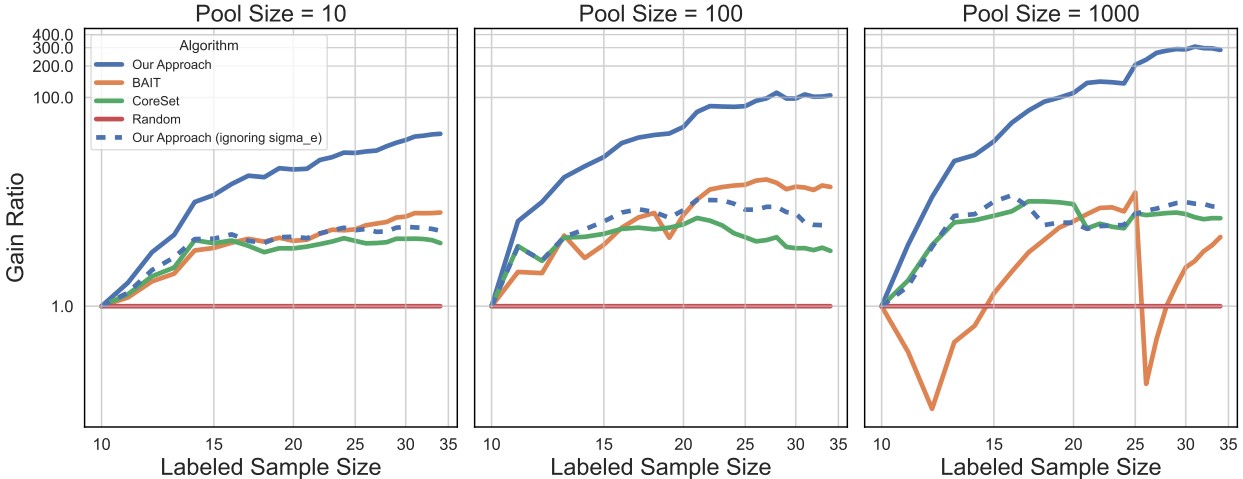

Figure 9: We repeat the experiment in Figure 8, but instead plot *Gain Ratio* (Equation 12) over $n = 2500$ realizations as a function of $|\mathcal{D}_L|$, the labeled sample size. In this plot, **higher is better**.

follows the measure of the top $k$ order statistics of the $\chi^2$ distribution with degrees of freedom equivalent to the number of parameters the user seeks to estimate. This is more formally expressed in Theorem 3.

**Theorem 3.** *Let $\mathcal{D}_P = \{\mathbf{x}_1, \ldots, \mathbf{x}_m\}$ be samples drawn from a normal distribution with mean zero and $s^2$ variance, i.e $\forall i \in [1, m]$. $\mathbf{x}_i \sim \mathcal{N}(0, s^2)$. Additionally, for each sample $\mathbf{x}_i$, let each component of $\mathbf{x}_i$ be drawn from a distribution with mean $0$ and variance $s^2$, i.e $\forall i \in [m]$. $\forall s \in [d]$. $x_{i,s} \sim \mathcal{N}(0, s^2)$. Let each sample be a d-dimensional vector, i.e $\forall i \in [1, m]$. $\mathbf{x}_i \in \mathbb{R}^d$. Let $\widetilde{\mathbf{X}} = \{\mathbf{x}_{(m-k-1)}, \ldots, \mathbf{x}_{(m)}\}$ be the $k$ points chosen under Algorithm 1 for sampling. Suppose without loss of generality, suppose the parameters of interest are the first $d'$ parameters, where $d' \leq d$. Then, the measure of $\widetilde{\mathbf{X}}$ is equivalent to the measure of the top $k$ order statistics of $\mathcal{D}_{sq} = \{y_1, \ldots y_m\}$, where $y_i = \sum_{j=1}^{d'} (\frac{x_{i,j}}{s})^2$. Furthermore, given $\mathcal{D}' = \{y_{(m-k+1)}, \ldots, y_{(m)}\}$ is the top $k$ order statistics of $\mathcal{D}_{sq}$,*

$$P(\mathcal{D}', m, k) = \frac{m!}{(m-k)!} \times \left[ \prod_{i=m-k+1}^{m} f_Y(y_{(i)}) \right] \left[ F_Y(y_{(m-k+1)}) \right]^{m-k} \tag{20}$$

*where $Y \sim \chi^2(df = d')$.*

Geometrically speaking, Theorem 3 essentially indicates that our sampling strategy selects points along a hypersphere where the radius reflects the information of the sample. This is reflected in Figure 10.

Figure 10(a) demonstrates that as pool sizes increases, our algorithm's selected points are further from the origin. Additionally, our sampling strategy selects points equidistant from the zero vector of the parameters we seek to estimate. In other words, this is a ball with radius that increases as pool size increases. However, as shown in figure 10(b), this trend is very sensitive to the presence of measurement error.

## 6 Beyond Linear Regression: Classification

We can apply our algorithm to Binary Classification Tasks, assuming we relax certain assumptions discussed further below. We formalize the univariate case below, but this can be easily extended to the multivariate case.

Suppose a logistic regression model with one independent variable, where $t = \text{logit}(p) = \alpha^\star + \beta^\star x$. Let $y \in \{0, 1\}$ be the binary outcome variable, where $p$ represents the probability of the positive outcome, $\alpha$ is the intercept term, and $\beta$ is the coefficient associated with the independent variable $x$.

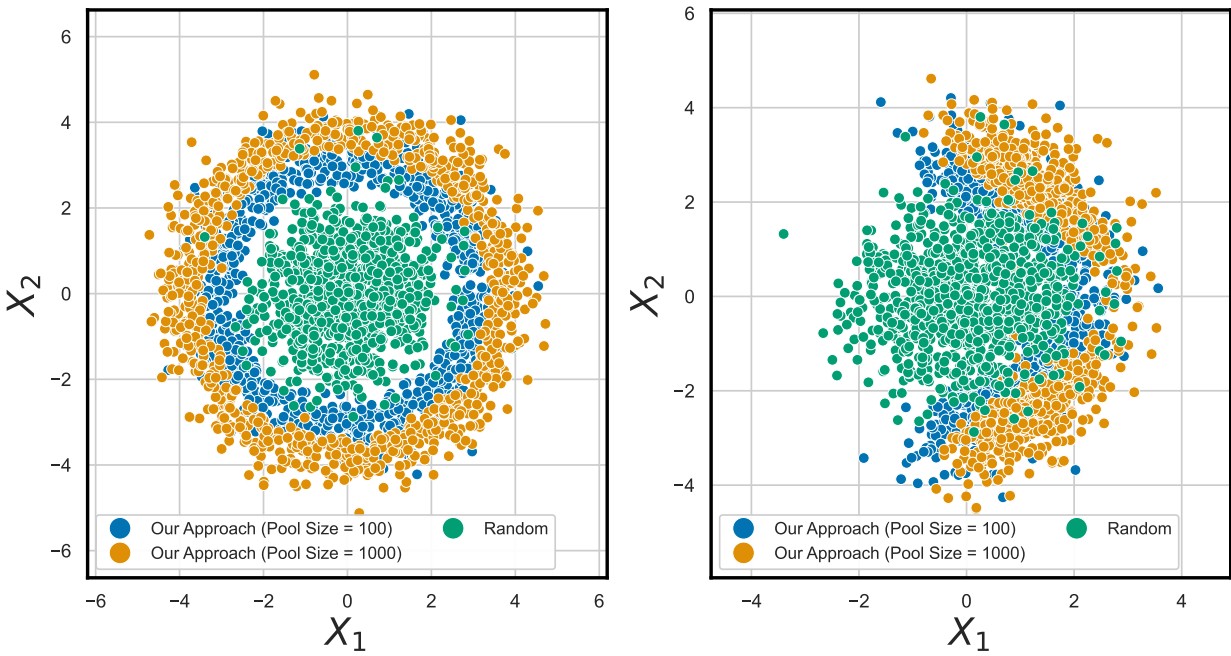

Figure 10: In both plots we sample and plot the $i = 1000$ additional points chosen by each algorithm, given $s = 10$ initial random datapoints. Given $\alpha^\star = 0$ and $\forall i.\ \beta_i^\star = 1$, we seek to estimate both $\beta_1^*$ and $\beta_2^*$. **Left:** The generator function is $y = \varepsilon + \beta_0^* x_0 + \beta_1^* x_1 + \beta_2^* x_2 + \beta_3^* x_3 + \beta_4^* x_4 + \alpha^\star$ (where no measurement error is present). **Right:** The generator function is $y = \varepsilon + \beta_0^* x_0 + \beta_1^* x_1 + \beta_2^* x_2 + \beta_3^* x_3 + \beta_4^* x_4 + \alpha^\star + \delta$ and the measurement error is $\delta \sim \mathcal{N}(0, \sigma_e^2)$ and $\sigma_e = \frac{(X_1 - 5)^2}{5}$.

## 6.1 Characteristics of the Adjusted Fisher Information

We utilize Equation 9 to compute the adjusted Fisher information. This case study allows us to study certain violations of our assumptions. Specifically, we acknowledge that two of our assumptions are violated in this example. First, we violate the assumption of homoskedasticity since the variance of the outcome at fixed $x$ is a function of $x$ itself. Second, we violate the normality assumption as the outcome likelihood does not follow a normal distribution. Thus, here we investigate whether our algorithm can still lead to improvements even when these assumptions are violated.

Suppose $t = \text{logit}(p) = \alpha^\star + \beta^\star x$. We have the estimated regression $\widehat{t} = \widehat{\beta} x + \widehat{\alpha}$, where $\widehat{\beta}$ and $\widehat{\alpha}$ are estimated quantities. Moreover, it is important to consider that the Fisher information for these parameters in this scenario is correlated. We shift $x$ to $x - \widehat{\mu}$, where $\widehat{\mu} = -(\widehat{\alpha}/\widehat{\beta})$ represents the midpoint of the logistic curve. We first perform this transformation and then compute the gradient with respect to the model parameters.

Thus, similar to the univariate linear regression case, manipulating the distribution of the labeled sample does not yield any additional information for $\alpha^\star$, but the most informative datapoints in the logistic setting for estimating $\beta^\star$ are at the tails of the input distribution.

## 6.2 Experimental Setup

Let $x \sim \mathcal{N}(\mu = 10,\ \sigma = 10)$ be an independent variable. Let the priors of the parameters be $\alpha \sim \mathcal{N}(\mu = 0,\ \sigma = 1)$ and $\beta \sim \mathcal{N}(\mu = 0,\ \sigma = 1)$. We draw posteriors using the No-U-Turn Sampler (NUTS), an adaptive Hamiltonian Monte Carlo sampling method (Hoffman et al., 2014). We study the potential improvement our algorithm provides when estimating $\alpha^\star$ or estimating $\beta^\star$ with respect to passive learning (random sampling).

### 6.3 Results

Figure 11 displays the adjusted Fisher information for both the intercept (blue curve) and slope (orange curve) after the decorrelation is applied. This visualization illustrates that in our active learning setting, where the primary focus is on learning the intercept, new data is sampled from where the predicted probability is near 50%. On the other hand, when the objective is to learn the slope, our active learning approach concentrates on areas where the predicted probability is approximately 18% or 82%.

In a binary classification task, most uncertainty-based approaches (Dagan & Engelson, 1995), rank events in the same order and pick the event with probability near 50%. These algorithms have been applied in numerous practical settings with varying degrees of success (Abernethy et al., 2018; Kennamer et al., 2020). Our analysis shows that these algorithms mostly focus on $\alpha^\star$; however, where the slope $\beta^\star$ is the quantity of interest, the uncertainty-based algorithms are suboptimal.

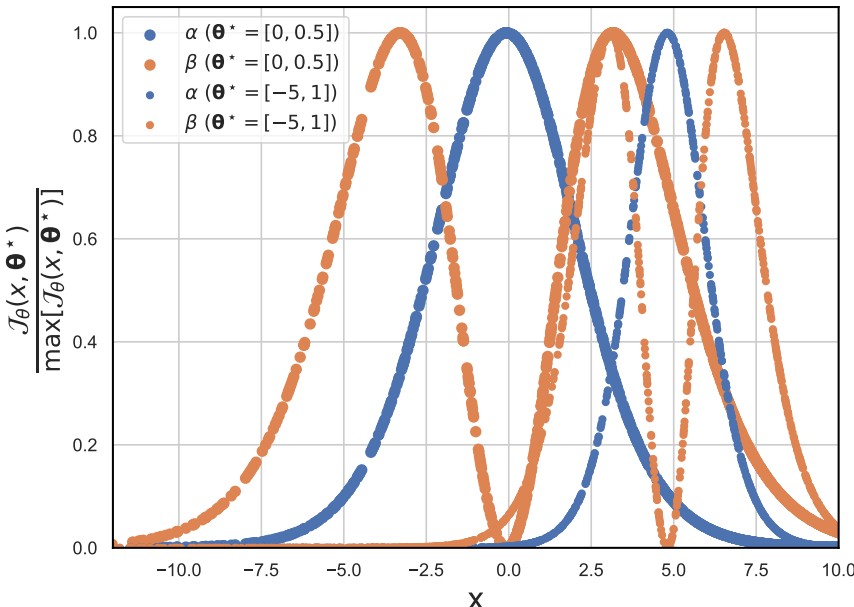

Figure 11: The adjusted Fisher information of the normalization ($\alpha^\star$) and slope ($\beta^\star$) for two set of parameter values: $\boldsymbol{\theta} = [0, 0.5]$ (big dots) and $\boldsymbol{\theta} = [-5, 1]$ (little dots).

## 7 Conclusion

Parameter estimation remains an important and relevant task. However, gathering labels for data points to train linear models can be expensive and time-consuming. Thus, we can leverage active learning techniques to select the points that are most helpful for estimating model parameters. Our proposed algorithm allows users to choose points that help best estimate parameters of interest, rather than reducing generalization error, and has been shown to be theoretically optimal in linear settings. Furthermore, unlike previous approaches, our solution allows users to account for measurement error. We thus demonstrate promising performance gains of our heuristic over other state-of-the-art approaches like BAIT and CoreSet.

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

## A   Appendix: Fisher Information Properties and Proofs

**Property 1** (Additive). *If data instances* $\mathbf{x}_i$ *and* $\mathbf{x}_j$ *are independent, then their conditional Fisher information is additive.*

$$\mathcal{I}(\{\mathbf{x}_i \cup \mathbf{x}_j\}, \theta) = \mathcal{I}(\mathbf{x}_i, \theta) + \mathcal{I}(\mathbf{x}_j, \theta).$$

*Proof.*

$$\mathcal{I}(\{\mathbf{x}_i \cup \mathbf{x}_j\}, \theta) = \mathbb{E}_{y_i, y_j} \left[ \frac{\partial}{\partial \theta} \ln\left(P(y_i, y_j \mid \mathbf{x}_i, \mathbf{x}_j, \theta)\right) \right]^2$$

$$= \mathbb{E}_{y_i, y_j} \left[ \frac{\partial}{\partial \theta} \ln\left(P(y_j \mid \mathbf{x}_j, \theta)\right) + \frac{\partial}{\partial \theta} \ln\left(P(y_i, \mid \mathbf{x}_i, \theta)\right) \right]^2$$

$$(\{\mathbf{x}_i, y_i\} \text{ and } \{\mathbf{x}_j, y_j\} \text{ are independent})$$

$$= \mathbb{E}_{y_j} \left[ \frac{\partial}{\partial \theta} \ln\left(P(y_j \mid \mathbf{x}_j, \theta)\right) \right]^2 + \mathbb{E}_{y_i} \left[ \frac{\partial}{\partial \theta} \ln\left(P(y_i, \mid \mathbf{x}_i, \theta)\right) \right]^2 \quad (\mathbb{E}_y \left[ \frac{\partial}{\partial \theta} \ln\left(P(y \mid \mathbf{x}, \theta)\right) \right] = 0)$$

$$= \mathcal{I}(\mathbf{x}_j, \theta) + \mathcal{I}(\mathbf{x}_i, \theta)$$

$\square$

**Corollary 1.** *Let* $\mathbf{x}_1, \ldots, \mathbf{x}_n$ *be i.i.d. draws from distribution* $q(\mathbf{x})$. *Then, under standard regularity conditions,*

$$\mathrm{Var}(\hat{\theta} - \theta^\star) \ \rightarrow \ \frac{1}{n\,\mathbb{E}_{\mathbf{x} \sim q}[\mathcal{I}(\mathbf{x}, \theta^\star)]}, \quad \text{as } n \rightarrow \infty.$$

*Proof.* Let $\{\mathbf{x}_1, \ldots, \mathbf{x}_n\}$ be i.i.d. draws from $q(\mathbf{x})$. By standard asymptotic normality of the maximum likelihood estimator,

$$\sqrt{n}(\hat{\theta} - \theta^\star) \ \xrightarrow{d} \ \mathcal{N}\left(0, \ \mathcal{I}_q(\theta^\star)^{-1}\right),$$

where

$$\mathcal{I}_q(\theta^\star) := \mathbb{E}_{\mathbf{x} \sim q}[\mathcal{I}(\mathbf{x}, \theta^\star)]$$

denotes the per-sample Fisher information under $q$. Therefore,

$$\mathrm{Var}(\hat{\theta} - \theta^\star) \ \rightarrow \ \frac{1}{n} \mathcal{I}_q(\theta^\star)^{-1}.$$

Equivalently, since Fisher information is additive over independent samples,

$$\mathcal{I}(\{\mathbf{x}_1, \ldots, \mathbf{x}_n\}, \theta^\star) = \sum_{i=1}^{n} \mathcal{I}(\mathbf{x}_i, \theta^\star) \ \rightarrow \ n\,\mathbb{E}_{\mathbf{x} \sim q}[\mathcal{I}(\mathbf{x}, \theta^\star)],$$

and hence

$$\mathrm{Var}(\hat{\theta} - \theta^\star) \ \rightarrow \ \frac{1}{\mathcal{I}(\{\mathbf{x}_1, \ldots, \mathbf{x}_n\}, \theta^\star)} = \frac{1}{n\,\mathbb{E}_{\mathbf{x} \sim q}[\mathcal{I}(\mathbf{x}, \theta^\star)]}.$$

$\square$

## B   Appendix: Theorem 2

**Theorem 2.** *Let* $\mathcal{D}_P = \{x_1, \ldots, x_m\}$ *be samples drawn from a normal distribution with mean zero and* $s^2$ *variance, i.e* $\forall i \in [1, m]$. $x_i \sim \mathcal{N}(0, s^2)$. *Let* $\widetilde{X} = \{x_{(m-k-1)}, \ldots, x_{(m)}\}$ *be the* $k$ *points chosen under Algorithm 1 for sampling. (Note:* $\widetilde{X} \subseteq \mathcal{D}_P$ *and* $x_{(i)}$ *is the ith order statistic and* $x_{(i)} \leq x_{(i+1)}$). *Then, the probability distribution of* $\widetilde{x}$ *is*

$$P(\widetilde{x}, m, k) = \frac{m!}{(m-k)!} \times \left[ \prod_{i=m-k+1}^{m} \frac{1}{s\sqrt{2\pi}} \exp\left(\frac{-(x_{(i)})^2}{2s^2}\right) \right] \times \left[ 2\Phi\left(\frac{|x_{(m-k+1)}|}{s}\right) - 1 \right]^{m-k} \tag{21}$$

*where* $\Phi\left(\frac{x}{s}\right) = \frac{1}{2}\left[1 + \mathrm{erf}(\frac{x}{s\sqrt{2}})\right]$ *is the CDF of* $\mathcal{N}(0, s^2)$.

*Proof.* – Since each $x_i$ is drawn from a distribution with mean 0, we can define a corresponding $z_i = \frac{x_i}{s}$, where $z_i$ is standard normal. We define $\mathcal{D}_{sq} = \{y_1, ..., y_m\}$, where $\forall i \in [1, m]$. $y_i = z_i^2 = \frac{x_i^2}{s^2}$. Choose an arbitrary $i$. Let $Z$ be the random variable representing $z_i$. Let $Y$ be the random variable representing $y_i = z_i^2$. It is known that $Y \sim \chi^2(df = 1)$.

Given $F_Y(y) = \frac{\gamma(\frac{1}{2}, \frac{y}{2})}{\Gamma(\frac{1}{2})}$,

$$
\begin{aligned}
f_Y(y) = \frac{d}{dy} F_Y(y) &= \frac{d}{dx} \left[ 2F_X\left(\frac{|x|}{s}\right) - 1 \right] \times \left| \frac{dx}{dy} \sqrt{y} \right| \\
&= \frac{d}{dx} \left[ 2F_X\left(\frac{|x|}{s}\right) - 1 \right] \times \frac{1}{2\sqrt{y}} \\
&= 2f_X\left(\frac{|x|}{s}\right) \times \frac{s}{2|x|} \qquad\qquad (\sqrt{y} = |z| = \tfrac{|x|}{s}) \\
&= \frac{s f_X(\frac{|x|}{s})}{|x|}.
\end{aligned}
$$

Thus, we may express $f_X\left(\frac{|x|}{s}\right) = \frac{|x|}{s} \times f_Y(y) = \sqrt{y} \times f_Y(y)$.

Algorithm 1 applied to linear regression will select the $k$ points with the highest $x_i^2$. This means the measure of the distribution of sampled points $\widetilde{X}$ is equivalent to the measure of the joint distribution of the top $k$ order statistics $\{y_{(m-k+1)}, \ldots, y_{(m)}\}$. Thus,

$$
\begin{aligned}
P(\widetilde{x}, m, k) &= P(\{\widetilde{x}_{(m-k+1)}, \ldots, \widetilde{x}_{(m)}\}, m, k) \\
&= \frac{m!}{(m-k)!} \times \left[ \prod_{i=m-k+1}^{m} \sqrt{y_{(i)}} \times f_Y(y_{(i)}) \right] \left[ F_Y(y_{(m-k+1)}) \right]^{m-k} \\
&= \frac{m!}{(m-k)!} \times \left[ \prod_{i=m-k+1}^{m} \sqrt{y_{(i)}} \times \frac{1}{\sqrt{2\pi}} (y_{(i)})^{-1/2} \exp\left(-\frac{y_{(i)}}{2}\right) \right] \left[ F_Y(y_{(m-k+1)}) \right]^{m-k} \\
&= \frac{m!}{(m-k)!} \times \left[ \prod_{i=m-k+1}^{m} \frac{1}{s\sqrt{2\pi}} \exp\left(\frac{-(x_{(i)})^2}{2s^2}\right) \right] \times \left[ 2\Phi\left(\frac{|x_{(m-k+1)}|}{s}\right) - 1 \right]^{m-k}
\end{aligned}
$$

When $k = 1$, the probability distribution simplifies to the following:

$$
P(\widetilde{x}, m, k) = \frac{m}{s\sqrt{2\pi}} \exp\left(\frac{-(\widetilde{x})^2}{2s^2}\right) \times \left[ 2\Phi\left(\frac{|\widetilde{x}|}{s}\right) - 1 \right]^{m-1}
$$

$\square$

We visualize the $k = 1$ case in Figure 12.

## C  Appendix: Theorem 3

**Theorem** 3. *Let $\mathcal{D}_P = \{\mathbf{x}_1, \ldots, \mathbf{x}_m\}$ be samples drawn from a normal distribution with mean zero and $s^2$ variance, i.e $\forall i \in [1, m]$. $\mathbf{x}_i \sim \mathcal{N}(0, s^2)$. Additionally, for each sample $\mathbf{x}_i$, let each component of $\mathbf{x}_i$ be drawn from a distribution with mean 0 and variance $s^2$, i.e $\forall i \in [m]$. $\forall s \in [d]$. $x_{i,s} \sim \mathcal{N}(0, s^2)$. Let each sample be a d-dimensional vector, i.e $\forall i \in [1, m]$. $\mathbf{x}_i \in \mathbb{R}^d$. Let $\widetilde{\mathbf{X}} = \{\mathbf{x}_{(m-k-1)}, \ldots, \mathbf{x}_{(m)}\}$ be the $k$ points chosen under Algorithm 1 for sampling. Without loss of generality, suppose the parameters of interest are the first $d'$ parameters, where $d' \leq d$. Then, the measure of $\widetilde{\mathbf{X}}$ is equivalent to the measure of the top $k$ order statistics of $\mathcal{D}_{sq} = \{y_1, ...y_m\}$, where $y_i = \sum_{j=1}^{d'} (\frac{x_{i,j}}{s})^2$. Furthermore, given $\mathcal{D}' = \{y_{(m-k+1)}, \ldots, y_{(m)}\}$ is the top $k$ order statistics of $\mathcal{D}_{sq}$,*

$$
P(\mathcal{D}', m, k) = \frac{m!}{(m-k)!} \times \left[ \prod_{i=m-k+1}^{m} f_Y(y_{(i)}) \right] \left[ F_Y(y_{(m-k+1)}) \right]^{m-k} \tag{22}
$$

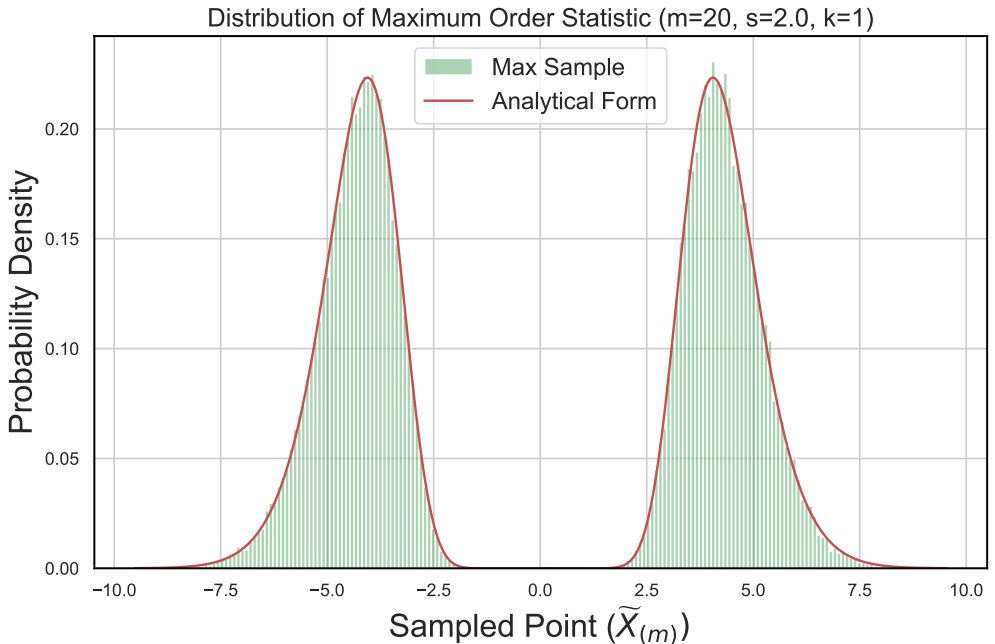

Figure 12: We demonstrate the simulated distribution of the point sampled under our algorithm matches our derived analytical form for the $k = 1$ case. The green histograms is an empirical distribution. In this exampl, we use a pool size $m = 20$ with draws from a normal distribution with mean zero and standard deviation of $s = 2$.

where $Y \sim \chi^2(df = d')$.

*Proof.* – For each chosen sample $\mathbf{x}_i$ we can define a corresponding set of random variables $z_{i,j} = \frac{x_{i,j}}{s}$, where $x_{i,j}$ is the $j$th component of $\mathbf{x}_i$. Choose an arbitrary $i$. Then $\forall j \in [d']$, let $Z_j$ be the standard normal random variable representing $z_{i,j}$.

Let $Y$ be the random variable representing $y_i = \sum\limits_{j=1}^{d'} (z_{i,j})^2$. Clearly, $Y \sim \chi^2(df = d')$.

Since Algorithm 1 essentially selects the top $k$ order statistics of $\mathcal{D}_{sq}$, we have

$$P(\mathcal{D}', m, k) = \frac{m!}{(m-k)!} \times \left[ \prod_{i=m-k+1}^{m} f_Y(y_{(i)}) \right] \left[ F_Y(y_{(m-k+1)}) \right]^{m-k}$$

$\square$

