# OpenReview forum: "Active Learning for Parameter Estimation in the Presence of Noise for Linear Models"
_TMLR — Rejected by TMLR_

### Review · Reviewer_CoAf · 2026-01-06

**Summary Of Contributions:**

The paper investigates active learning using the expected information gain for linear models with homoscedastic noise. It compares to modern approaches, that is CoreSet and BAIT, on toy settings (1D and 5D).

Its approach is based on using the Fisher Information and focuses on univariate regression mainly with an extension to multivariate linear models and logistic regression.

**Audience:**

Yes

**Audience Explanation:**

Assuming my concerns are resolved, this paper could be an interesting tutorial paper at TMLR and a useful introduction to Active Learning using Fisher Information.

**Broader Impact Concerns:**

None.

**Claims And Evidence:**

No

**Claims Explanation:**

Starting with the positives, the paper is clearly written and tries to focus and constrain its contributions. The results are a useful collection of theory for the active learning of univariate linear models under noise. In particular, the separation of noise into model noise (which is estimated) and measurement noise is neat.

At the same time, there are some potential inaccuracies that require clarification by the authors.

Firstly, the title: noisy labels seems a bit of a misnomer given the focus on linear regression. Why not “Data Valuation in the Presence of Noise for Linear Models”? At the same time: why data valuation? The described setup is clearly an active learning setup (as also evidenced by the related citations).

Secondly, the literature review, while extensive on the surface, misses some incredibly relevant literature. Linear models have been around for a long time and so has active learning. §2 mentions experimental design but does not delve (sorry!) deeper. Starting with Fedorov’s "Theory of Optimal Experiments" (1972) to more recent works (I’m sorry I’m not well-versed with this part myself), but a comparison to D-, A-, E-, c-optimality (see e.g. https://en.wikipedia.org/wiki/Optimal_experimental_design) would be great and necessary.

The proposed method seems similar to A- or c-optimality. Similarly, while there is no claim to be Bayesian, the approach is equivalent to maximizing the expected information gain over x:

$$\max I[Y;W|x] = \max H[W] - H[W|Y,x] = \min H[W|Y,x] = \min Var[W|Y,x] = \min I[W|x]^{-1} = \max I[W|x],$$

where I used $I[W|x]$ for the conditional Fisher Information, and we can drop $H[W]$ because it is independent of $x$.

There are also several papers by David MacKay from the 1990s that look into this and might also be worth comparing to. E.g. MacKay, David JC. "Information-based objective functions for active data selection." (1992) and “The evidence framework applied to classification networks.” (1992).

More recently, "Unifying Approaches in Active Learning and Active Sampling via Fisher Information and Information-Theoretic Quantities." (Kirsch & Gal, TMLR) looks into the connection of Fisher Information for weight/parameter-space AL methods and prediction-space AL methods (e.g. BALD) and the original information-theoretic quantities (Expected Information Gain). In this context, it also shows that the proposed method in this paper is equivalent to the EIG and also BALD. (This is more of a tutorial paper.)

This brings me to another point: the paper claims that active learning is inference-oriented and not parameter-oriented. However, in general, this is not true. This is the point of "Prediction-oriented Bayesian Active Learning." (Smith, Freddie Bickford, et al., AISTATS 2023): BALD and other methods specifically target the parameter EIG and are not prediction-oriented. (BAIT is prediction-oriented however as are other transductive methods or error-reduction-focused methods.)

Thus, the novelty claim in the contributions do not hold in my opinion. This is not a reason for rejection at TMLR but requires clarification by the authors before I can agree that the claims are all evidenced.

Now on to §3:

I believe Assumption 2 does not make sense. In particular, it is not clear how parameters can be “linearly independent from each other”. I think this might refer to the Fisher Information being a diagonal matrix for those parameters, but this is not feasible because the Fisher information for a point x is $x x^T$ and cannot be diagonal, so this can only hold for univariate linear models (and bias). This would need to be clarified as it limits the applicability of the theory and derivations a lot.

Now on to §4.1.3:

The multivariate case is not well-motivated. For one, the unnumbered equation is wrong. It should prob be $tr(\nabla ... \nabla ...^T)$ but it lacks the second part. Eq (10) is correct. The use of the trace is not motivated, however. More principled approaches use the determinant, but see also A- vs D-optimality.

Furthermore, the presented algorithm is *not* batch-aware contrary to the claims in the paper. This is also explained in “Unifying Approaches…” by Kirsch & Gal, but the trace formulation cannot capture correlations between points because it is linear.

The easiest way to see that in the presented algorithm is to assume you have a 1000 points with high contribution to parameter 1 and 1000 points with lower contribution to parameter 2. The algorithm will pick all the points with contributions to parameter 1 and ignore the ones that contribute to parameter 2, even though it would be beneficial to select from both.

This is a lot already. I’ll await the author's initial response before continuing.

**Requested Changes:**

See above.

---

### Review · Reviewer_HJdM · 2026-01-13

**Summary Of Contributions:**

The paper proposes a Fisher-information-based strategy for data valuation and acquisition when the goal is parameter estimation rather than predictive performance. In each acquisition step, the method assigns each unlabelled sample an adjusted information score. This score is approximated with respect to a user-specified subset of parameters, and the algorithm then acquires k points with the highest scores. Compared with standard acquisition rules that primarily optimise predictive accuracy, this criterion is explicitly designed to improve inferential precision for the parameters of interest. The paper provides theoretical results for univariate and multivariate linear regression with measurement error. It then follows with synthetic experiments and an extension to logistic regression.

**Strengths**
- The focus on parameters of interest rather than predictive accuracy is well-motivated for some applications.
- The proposed acquisition rule is computationally simple and derived based on properties of conditional Fisher information.
- The paper incorporates known measurement-error variance into the acquisition rule and proposes a gain-ratio style evaluation.

**Weaknesses** (details below)
- There appear to be multiple mathematical errors in their claims.
- The novelty relative to classical optimal design and Fisher-information-based active learning is less clear.
- The empirical section has several gaps in terms of its interpretability and justification.

**Audience:**

Yes

**Audience Explanation:**

The paper addresses active learning and subset selection for parameter estimation. In addition, it explicitly incorporates measurement error and heteroscedasticity for the problem setup. This work can be an interest to communities specialised in active learning, experimental design, statistical ML, and scientific ML. The idea for prioritising data that reduces uncertainty for a targeted estimand $\theta_s$ is well aligned with classical design criteria and could be useful if the theory is corrected and the novelty is positioned carefully.

**Broader Impact Concerns:**

No concerns. The work is primarily methodological/theoretical.

**Claims And Evidence:**

No

**Claims Explanation:**

Several claims are not written accurately or precisely. (Please correct me if I misunderstood anything.)
1. The theoretical justification relies on parameters being decoupled, which is described as having negligible off-diagonal Fisher information terms. However, the formal condition in assumption 2 is stated as "linear independence", which does not imply the decoupling condition.
2. In Eq. (5), shouldn’t the asymptotic variance scale like $\mathrm{Var}(\hat \theta - \theta^\ast)\rightarrow \frac{1}{n\mathbb{E}_{q(x)}[\mathcal I (x,\theta^\ast)]}$? This follows directly from the definitions of conditional, marginal Fisher information and standard asymptotic normality results (cf. Eq. (1)–(3)).
3. The top equation in Section 4.1.3 appears to take a trace of a vector $\nabla_\theta \ln P(\cdot)$, and this is not well-defined.
4. Eq. (11) seems to use incorrect distributions and contradicts the definition.
5. In logistic regression setting, the decorrelation shift $x\rightarrow x-\alpha/\beta$ depends on the true unknown parameters, which is not viable. I also found the claim $f_\theta(x)=\hat \beta (x-\alpha/\beta)+\alpha$ is confusing with the notation, since $f_\theta$ is presented as a parametric function while the expression mixes hatted/unhatted symbols.

**Requested Changes:**

**Correctness, theory, and notation**
- Fix the incorrect or inconsistent statements, including those in the second part of this review.
- Make notation consistent (e.g., distributions and symbols in Eq. (11) and Eq. (16)).
- Fix the derivation in Section 5.1.1. In particular, after the last equality, there is no $\theta$-dependence, thus $|_{\theta=\hat\theta}$ is redundant.
- Correct missing/incorrect citations. For example, the citation formatting in Section 2.1 (Influence functions) appears wrong, and the final sentence of that paragraph should cite the specific recent work being referenced.

**Problem setup**
- Justify more about the acquisition setup. This method assumes that each round draws a new pool of size m and discards m-k points, which is reasonable in streaming settings but deviates from the standard pool-based active learning where the unlabeled set remains the same. The paper should explain how to adapt this in the standard setting and whether the theory changes.

**Experiments and empirical justification**
- Improve interpretability and align the empirical story with the theory. For example: (i) Explain what is being demonstrated by the “quadratic-in-ln” fits in Fig. 2 and why that functional form is expected. (ii) Clarify the interpretation of uneven peaks in Fig. 5 and connect them to the experimental setup.
- Consider adding ablations, e.g., (i) without the measurement-error adjustment, (ii) with true vs estimated $\sigma^2_e(x)$, (iii) with/without centering/reparameterisation.

**Positioning and novelty relative to prior work**
- Clarify novelty relative to classical optimal experimental design and active regression. Many results, such as selecting extremes for slope and inverse-variance weighting under heteroscedasticity, are closely related to established design principles [1]. The paper would benefit from explicitly positioning the method as a computationally simple approximation to a standard design criterion for parameter subsets, and clarifying what is new (e.g., streaming pool framing, specific measurement-error integration, implementation advantages).


[1] Pukelsheim, Friedrich. Optimal design of experiments. Society for Industrial and Applied Mathematics, 2006.

---

### Review · Reviewer_Afup · 2026-01-18

**Summary Of Contributions:**

The paper introduces a data valuation framework designed for precise parameter estimation rather than global predictive accuracy. The authors present a parameter-focused metric utilizing a Fisher-information-based criterion to identify the most informative samples. They demonstrate that, assuming asymptotic normality of the posterior, this strategy is mathematically equivalent to minimizing the variance of the target parameter estimates. To ensure robustness, the authors introduce an adjusted Fisher Information metric that incorporates measurement error and heteroscedasticity, allowing for data selection in more realistic settings. Moreover, the authors show convergence in linear models and generalized linear models with logit link function (logistic regression). Their analysis also establishes that the gain ratio, which represents the performance advantage over random sampling, scales logarithmically with the unlabeled pool size. The resulting selection algorithm leverages the additivity of conditional Fisher information to pick the top $k$ samples from a pool of size $m$, achieving $O(m \log m)$ scaling which makes it suitable for large datasets.

**Audience:**

Yes

**Audience Explanation:**

Yes, the paper is relevant to the TMLR audience. While the current scope focuses on fundamental models (linear and logistic), the methodology provides a principled approach to data valuation for specific inferential goals. This makes the work specifically relevant to the growing community of researchers at the intersection of machine learning and other scientific disciplines.

**Claims And Evidence:**

Yes

**Claims Explanation:**

The authors support their claims with both theoretical and empirical evidence. They provide closed-form expressions for information gain and demonstrate for linear models that the optimal sampling strategy is inherently parameter-specific. For instance, they show that minimizing the variance of a slope estimate requires sampling from the tails of the covariate distribution, whereas the intercept is better estimated using samples near the center. Their analysis also establishes that the gain ratio, i.e. the resulting performance advantage over random sampling, scales logarithmically with the unlabeled pool size. Furthermore, empirical results demonstrate that the proposed method consistently achieves comparable or superior convergence and higher gain ratios compared to BAIT and CoreSet.

**Requested Changes:**

No changes requested.

---

### Decision · Action_Editor_QHgL · 2026-03-11

**Recommendation:** Reject

**Additional Comments:**

In their final recommendations one reviewer recommended acceptance, one was leaning accept, and one was leaning reject.  All had concerns relating to the novelty of the work and the empirical evaluations, while two had issues in the presentation of the work and whether claims were fully justified; there were differences in how reviewers felt these issues lined up compared to the TMLR acceptance criteria.  Having read the work myself, my opinion is more in line with negative assessments, in fact I think I am probably more negative in my own assessment than any of the reviewers were (see earlier comments in this meta review).

Taking into both my own opinions and those of the reviewers, my overall recommendation is that the paper falls below the acceptance threshold, albeit relatively narrowly so.  My single biggest concern relates to the pitching of the work relative to classical OED approaches, and I do not feel the work is publishable without more clearly positioning itself in context of such classical work, with clearer and more explicitly justified claims in how it differs.  I thus encourage the authors to return to the work to try and address this and the other points I have raised above, but do not believe the paper is suitable for publication in its current form.

**Audience:**

Yes

**Audience Explanation:**

I believe the question of whether there will be interest from TMLR's audience is a borderline one, but I am willing to go with the consesus opinion from the reviewers here.  My feeling is that the differences in the approach to classical OED (especially c-optimality) are somewhat nominal and the restriction to linear models means the audience for the paper is going to be quite limited.  The poor clarity of the work also detracts from the potential interest.  However, I think some of the theoretical analysis is genuinely new and may be of interest to some readers, hence my ultimate decision of passing this criteria.

**Claims And Evidence:**

No

**Claims Explanation:**

I believe that the paper falls below TMLR's bar for supporting evidence in a number of critical ways:
- The empirical evaluations are very basic and do not test the approach in settings that are commonly used in practice.  Yes linear models are a common and important approach, but there is also a huge precedence for these in the classical optimal experimental design (OED) literature and there is no demonstration that the suggested approach improves over other classical setups that are also explicitly design for linear models.  In short, comparisons to methods like BAIT and CoreSet would be interesting when using modern machine learning methods, and comparisons to other classical OED approaches would be interesting for the linear models (e.g. comparing to D-optimality), but the paper actually has neither.
- I feel the paper rather mischaracterises previously work, especially the classical OED literature.  It is natural and well established in classical OED to only target information in the parameters you care about; this is essentially exactly what c-optimality is (though the setting here is slightly more restricted as all weights have to be 0 or 1).  Working in the streaming setting also has no real bearing on the approach, changing only the candidate points.  In my opinion, the paper thus claims novelty on aspects where I do not think there is any meaningful novelty.  Discussions on existing active learning work are often also misleading, with others also pointing out the need to target information in particular variables of interest (e.g. EPIG) and the distinctions between Fisher and Shannon information never really explained and the two often unhelpfully conflated.
- Though big improvements appear to have been made in the update, there are still various technical question marks hanging over the paper.  For example, proposition 1 appears to lack any proof.  Assumption 2 is not really justified as reasonable and later parts of the paper appear not to have been appropriately adapted to account for it.  The new claim of equivalence to EIG/BALD is misleading as this only holds for linear regression with a Gaussian likelihood and breaks (I believe) even once you move to logistic regression.  I generally feel another round of revising the paper and then re-reviewing is needed here.
- The clarity of the work is poor: though the low level writing is generally fine, the paper is bloated and does not have a clear flowing narrative, meandering between different points instead of focusing on clearly explaining the core contributions.  Lots of assumptions are made and it is not always clear what their implications are and how restrictive the setup is.  It is also not generally made clear what is being done that is different to standard Fisher information based experimental design, with the distinction between explaining well known results and potentially novel contributions often blurred.

**Resubmission Of Major Revision:**

The authors may consider submitting a major revision at a later time.